# Near Optimal Robust Federated Learning Against Data Poisoning Attack

**Jingfan Yu**
Tsinghua University
Shanghai Qi Zhi Institute
yujf20@mails.tsinghua.edu.cn

**Zhixuan Fang**[*]
Tsinghua University
Shanghai Qi Zhi Institute
zfang@mail.tsinghua.edu.cn

## Abstract

We revisit data poisoning attacks in the federated learning system. Consider $m$ worker nodes (each has $n$ training data samples) cooperatively training one model for a machine-learning task, and a fraction (i.e., $\alpha$) of the workers may suffer from the data poisoning attack. We mainly focus on the challenging and practical case where $n$ is small and $m$ is large, such that each worker does not have enough statistical information to identify the poisoned data by itself, while in total they have enough data to learn the task if the poisoned data are detected. Therefore, we propose a mechanism for workers to cooperatively detect workers with poisoned data. In terms of attack loss, our mechanism achieves $\tilde{O}((\frac{1}{n})^{\frac{1}{2}} + (\frac{d}{mn})^{\frac{1}{2}})$ in IID setting and $\tilde{O}((\frac{1}{\gamma})^{\frac{1}{2}} + (\frac{1}{n})^{\frac{1}{2}} + (\frac{d}{mn})^{\frac{1}{2}})$ in non-IID setting, where $d$ is the VC-dimension of the learning model and $\gamma$ is a concentration parameter characterizing the non-IIDness. Alongside attack loss, our mechanism limits the adversary's free-ride gain even when it cannot be directly quantified by the attack loss. We also propose the lower bound of the attack loss, and our proposed algorithm matches the lower bound when $m \to \infty$ both in IID setting and non-IID setting.

## 1 Introduction

Federated learning trains a target model across distributed nodes without centralizing data. A server dispatches the target model, workers return gradients, and the server aggregates them to update the model. Federated learning inherits the vulnerabilities to poisoning attacks. There are two types of poisoning attacks: model poisoning Chen et al. (2017); Su & Xu (2018); Yin et al. (2018); Blanchard et al. (2017) and data poisoning Sun et al. (2021); Kairouz et al. (2021); Tolpegin et al. (2020). Model poisoning attack refers to the attack where malicious workers modify the gradient sent to the server. These malicious workers are completely untrustworthy.

While model poisoning attack attracts much attention in the literature, there is a rising concern about a weaker yet more practical version of the poisoning attack, the data poisoning attack. Data poisoning attack refers to an attack where an adversary modifies the dataset of corrupted workers, as in Figure 1. The corrupted workers are honest, but their raw data is unknowingly corrupted by the adversary. Thus, compared to the model poisoning attack, the gradient of workers reflects their own dataset, even when the dataset is contaminated. For example, data poisoning allows a non-expert attacker to implement the classic and effective attack of label flipping without knowing the target model type or controlling the workers.

Yet little attention has been paid to data poisoning. Literature usually adopts the gradient robust aggregation method as the primary means of countering adversarial attacks Kairouz et al. (2021). The gradient robust aggregation method, originally designed to counter the model poisoning attack, also mitigates the data poisoning attack. However, since the data poisoning attack is a weaker attack, more efficient defenses for the data poisoning attack should exist.

Our research question is how well we can defend against arbitrary data poisoning in the worst case. The defense is characterized by two objectives: maximizing the target model accuracy of the unpoisoned data and mitigating the adversary's gain from the attack. The two objectives are not equivalent

---

[*]Corresponding author.

because there exists the so-called Trojan attack: The adversary can make the trained target model succeed in both meeting the attacker's goal and achieving high accuracy among the unpoisoned data.

Our solution in this paper addresses this question in the following ways.

**(a) We establish the minimax lower bound of the target model attack loss due to the poisoning attack.** Following the previous literature Yin et al. (2018); Blanchard et al. (2017), we focus on the case where more than half of the datasets are unpoisoned. The detection difficulty increases as the unpoisoned datasets' correlation decreases. Therefore, we develop a quantitative relation between the attack loss lower bound and the correlation. More specifically, in terms of how the workers' data are sampled, we consider the IID setting and the non-IID setting. In the IID setting, each worker samples $n$ data samples independently from the same distribution. In the non-IID setting, the Dirichlet distribution with concentration parameter $\gamma$ is used to characterize the non-IID extent Hsu et al. (2019). We show that the lower bound of attack loss is $\Omega(\frac{1}{\sqrt{n}})$ in IID setting and $\Omega(\frac{1}{\sqrt{\gamma}} + \frac{1}{\sqrt{n}})$ in non-IID setting. The parameters $n$ and $\gamma$ characterize the dataset similarity. More details will be in Section 3.

**(b) We propose an algorithm to asymptotically match the lower bound.** The challenge is how to discriminate between poisoned and unpoisoned datasets. Little literature works on data poisoning, researchers believe that the robust gradient aggregation defense against model poisoning attack can also address data poisoning attack Kairouz et al. (2021). Robust gradient aggregation relies on examining the gradients that the workers compute on the target model and removing outliers as poisoned ones. However, direct application of this method to the scenario of the data poisoning attack will introduce two limitations. First, the server needs to distinguish the malicious gradient every time it collects gradients, inducing continuous computational overhead. Second, the gradient may not faithfully reflect whether the corresponding dataset is poisoned. The gradient is a high-dimensional vector, and outlier detection is more difficult with higher-dimensional vectors. Figure 1(b-c) shows the two principal axes of gradients calculated by the unpoisoned datasets and a poisoned dataset, where the attack is the label-flipping attack. The gradients in the two figures are calculated from the same datasets, while the target model size is different. When the model is larger (Figure 1(c)), the gradient dimension is larger and the poisoned one is harder to detect. In addition, in the data poisoning attack, the gradient is generated according to the poisoned data instead of being arbitrarily generated. Such patterns of poisoned gradients are not utilized in robust gradient aggregation.

To solve this challenge, we train a discriminator model to directly differentiate among the workers' datasets instead of differentiating the gradients. To train the discriminator model, we propose a variance notion that characterizes the variances of the datasets as the training loss function. Our variance notion is also aligned with the definition of $\mathcal{H}$-divergence, which describes the distance between the datasets. To train the discriminator model, workers send gradients and model outputs to the server so that the server does not need direct access to the datasets. In our algorithm, the discriminator model is trained before starting the training of the target model. Compared with the gradient robust aggregation method, our algorithm is more effective and tailored to the data poisoning attack scenario by exploiting the feature that the gradient of each worker faithfully reflects its dataset (whether the dataset is poisoned or not).

Our algorithm achieve $\tilde{O}((\frac{1}{n})^{\frac{1}{2}} + (\frac{d}{mn})^{\frac{1}{2}})$ attack loss in IID setting and $\tilde{O}((\frac{1}{\gamma})^{\frac{1}{2}} + (\frac{1}{n})^{\frac{1}{2}} + (\frac{d}{mn})^{\frac{1}{2}})$ in non-IID setting, where $m$ is the number of workers and $d$ is VC-dimension of the learning task. Previous literature achieves the attack loss of $O((\frac{trace(\Sigma_g)}{n})^{\frac{1}{2}} + (\frac{d_g}{mn})^{\frac{1}{2}})$ Feng et al. (2014); Yin et al. (2018) and $O((\frac{\|\Sigma_g\|_2}{n})^{\frac{1}{2}} + (\frac{d_g}{mn})^{\frac{1}{2}})$ Su & Xu (2018) in IID setting, where $\Sigma_g$ is the covariance matrix of gradients and $d_g$ is the dimension of the gradients. Since $trace(\Sigma_g)$ and $\|\Sigma_g\|_2$ increases with the size of models Karakida et al. (2021), our algorithm achieves a better bound. When $m \to \infty$, the upper bound produced by our algorithm matches the lower bound.

**(c) We propose a notion that characterizes the adversary's gain from the attack. We show that our algorithm mitigates the adversary's gain both theoretically and experimentally.** To mitigate the attacker's gain from free ride or Trojan attacks, we propose a robustness metric, effective poison rate (EPR), for data poisoning attacks in federated learning systems, alongside the attack loss. EPR of a specific algorithm reflects the fraction of data that is effectively poisoned. For example, EPR of $c$ indicates that the training outcome is the same as a locally trained model with a proportion $c$ of the data being poisoned. If an algorithm achieves $c < \alpha$, where $\alpha$ is the proportion of workers that

the adversary can corrupt, the attack influence is sub-proportional and thus reduced. Our proposed mechanism achieves an EPR bound of $\tilde{O}((\frac{1}{n})^{\frac{1}{2}} + (\frac{d}{mn})^{\frac{1}{2}})$ in IID setting and $\tilde{O}((\frac{1}{\gamma})^{\frac{1}{2}} + (\frac{1}{n})^{\frac{1}{2}} + (\frac{d}{mn})^{\frac{1}{2}})$ in non-IID setting, demonstrating that, with sufficient samples, the adversary's impact can be effectively constrained.

Our theoretical contribution focuses on the region where the ratio of poisoned datasets $\alpha < \frac{1}{3}$, considers the optimization of model parameters as a black box, and assumes that we can get an optimized parameter from the black box. In our experiment, we use stochastic gradients to optimize the parameters and show that the optimization can be done with reasonable computation costs.

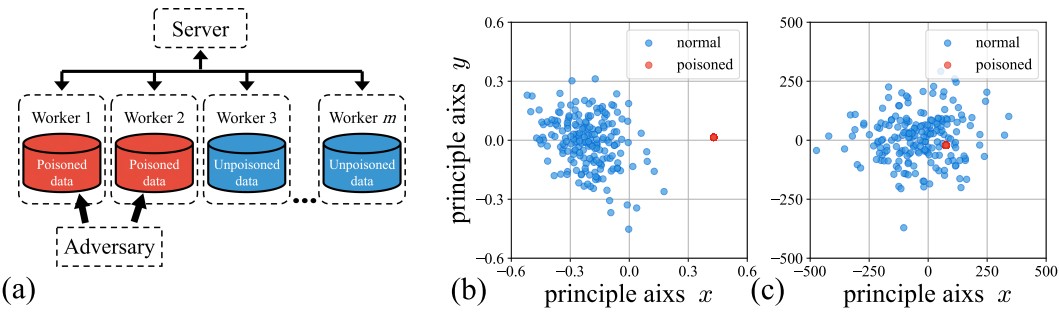

(a)  (b)  (c)

Figure 1: (a) Data poisoning attack in a federated learning system. (b-c) Two principal axes of gradients calculated by the same datasets on different neural networks. (b) The gradients are computed from a dense neural network (DNN) with 2 hidden layers and 128 neurons each hidden layer. (c) The gradients are computed from a DNN with 32 hidden layers and 128 neurons each hidden layer.

## 2 PROBLEM FORMULATION AND GOALS

**Participant.** We consider the federated learning setting with one parameter server and $m$ workers. Each worker $i$ has $n$ data points as his local dataset $\mathcal{W}_i$.

**Data space.** Let $\mathcal{Z} := \mathcal{X} \times \mathcal{Y}$ be the data space, where $\mathcal{X}$ is the set of objects that we may wish to label and $\mathcal{Y} = \{-1, 1\}$ is the label set. The binary-label setting can be generalized to the multi-label setting by combining the original object and the label into a single composite object, then assigning a binary label that indicates whether the label is the original label Mirza & Osindero (2014). Therefore, we focus on the binary-label setting in the theoretical part.

**Attack model.** An adversary $\mathcal{A}$ can corrupt the dataset of $\alpha$ fraction of workers and arbitrarily poison these datasets. Let $S_p \subseteq [m]$ be the set of workers with poisoned data and $S_n = [m] \setminus S_p$ be the set of normal workers. Therefore, $|S_p| = \alpha m$ and $|S_n| = (1 - \alpha)m$. The adversary $\mathcal{A}$ completely decides the dataset of workers in $S_p$, while it can not control subsequent actions of the workers. Also, the adversary cannot manipulate the datasets of normal workers.

**Normal worker dataset.** In terms of the distribution of local datasets of normal workers, we consider an IID setting and a non-IID setting. Let $\mathcal{P}(\mathcal{Z})$ and $\mathcal{P}(\mathcal{Z}^n)$ be the set of distributions over data points and $n$-sample datasets. In general, the dataset of each normal worker is sampled from a distribution $P \in \mathcal{P}(\mathcal{Z}^n)$. For the IID setting, each data point of the normal worker is sampled independently from an unknown ground truth distribution $D \in \mathcal{P}(\mathcal{Z})$. This means that the dataset of each normal worker is sampled from a distribution $D^{\otimes n} \in \mathcal{P}(\mathcal{Z}^n)$. For the non-IID setting, Dirichlet distribution is used to characterize the non-identicalness detailed in Section 3.2.

**Learning model** To carry out learning tasks, we consider a set of parameterized learning models $\{g_\theta\}_{\theta \in \Theta}$, where $\Theta$ is the parameter space and each $g_\theta$ is a function from $\mathcal{X}$ to $\mathbb{R}$. The set of learning models can induce a hypothesis class $\mathcal{H} = \{sign \circ g_\theta | \theta \in \Theta\}$. Each hypothesis is a map from $\mathcal{X}$ to $\mathcal{Y}$. Here, each hypothesis is a composition of a sign function and the learning model function. Let $d$ denote the VC-dimension of $\mathcal{H}$ Shalev-Shwartz & Ben-David (2014).

Let $\mathcal{M}$ be a federated learning mechanism. Mechanism $\mathcal{M}$ takes $m$ datasets as input and outputs a hypothesis, denoted as $\mathcal{M}(\{\mathcal{W}_i\}_{i=1}^m) = h$.

To measure the success of the trained target model, we use the error of the output hypothesis over the ground truth distribution $D \in \mathcal{P}(\mathcal{Z})$. For each $h \in \mathcal{H}$, the error is defined as follows:

$$\epsilon_D(h) = \mathbb{P}_{(x,y)\sim D}[h(x) \neq y] \tag{1}$$

## 2.1 ATTACK LOSS

For a mechanism $\mathcal{M}$ that outputs $h$ under attack, we define the attack loss to be:

$$R_D(h) = \epsilon_D(h) - \min_{h'\in\mathcal{H}} \epsilon_D(h') \tag{2}$$

The attack loss characterizes the increment of the target-model error incurred under attack relative to the attack-free case.

## 2.2 EFFECTIVE POISON RATE (EPR)

In addition to the attack loss, we also have a robustness goal aimed at limiting the effectiveness of the adversary $\mathcal{A}$. We define a metric EPR in Definition 2.2 that characterizes the effectiveness based on the concept of $\mathcal{H}$-divergence in Definition 2.1.

$\mathcal{H}$-divergence Ben-David et al. (2010) is modified from the total variation distance of two distributions to measure the difference related to the hypothesis class $\mathcal{H}$. The definition of $\mathcal{H}$-divergence is as follows:

**Definition 2.1** ($\mathcal{H}$-divergence). The $\mathcal{H}$-divergence between the two distributions $D$ and $D'$ for label $y$ is

$$d_{\mathcal{H}}(D, D') = \sup_{h\in\mathcal{H},\mathbf{a}\in\triangle^1} \left| \sum_{y'\in\mathcal{Y}} a_{y'} \left( \mathbb{P}_{(x,y)\sim D}[h(x) = 1, y = y'] - \mathbb{P}_{(x,y)\sim D'}[h(x) = 1, y = y'] \right) \right| \tag{3}$$

, where $\mathbb{P}$ represents the probability, $(x, y) \sim D$ denotes $(x, y)$ sampled from distribution $D$, $\mathbf{a} = \{a_{y'}\}_{y'\in\mathcal{Y}}$ is a 2-dimension probability vector, and $\triangle^1$ is the corresponding probability simplex.

The intuition is that since a hypothesis divides the input space $\mathcal{X}$ into two subsets, either being $h(x) = 1$ or 0, $\mathcal{H}$-divergence $d_{\mathcal{H}}$ specifies the best hypothesis $h$ that can be used to differentiate the samples with label $y'$ generated from two distribution $D$ and $D'$. When $\mathcal{H}$-divergence is 0, it means no hypothesis can differentiate the two distributions.

We now define EPR that characterizes the effectiveness of an adversary $\mathcal{A}$.

**Definition 2.2** (EPR). A federated learning mechanism $\mathcal{M}$ is $c$-EPR against an adversary that controls $\{\mathcal{W}_i\}_{i\in S_p}$, if there exists a dataset $\mathcal{D}$ that satisfies the followings:

1. Dataset $\mathcal{D}$ is close to the normal workers' dataset, with a distance at most $c$ in terms of $H$-divergence: $d_{\mathcal{H}}(\cup_{i\in S_n}\mathcal{W}_i, \mathcal{D}) \leq c$.

2. The output of $\mathcal{M}$ is the same in two cases: (a) the data set of each worker $i$ is $\mathcal{W}_i$, (b) the data set of each worker is all $\mathcal{D}$. Write as $\forall S \subseteq \mathcal{H}$, $\mathbb{P}[\mathcal{M}(\{\mathcal{W}_i\}_{i\in[m]}) \in S] = \mathbb{P}[\mathcal{M}(\{\mathcal{D}\}_{i\in[m]}) \in S]$.

We slightly abuse the notion that for a dataset $\mathcal{D}$, $x \sim \mathcal{D}$ denote $x$ is uniformly sampled from $\mathcal{D}$.

Roughly speaking, for a mechanism with $c$-EPR, we guarantee that the result of the training is equivalent to training a model on a dataset where no more than a $c$ proportion of total data is poisoned. A direct learning mechanism on datasets $\{\mathcal{W}_i\}_{i\in[m]}$, such as FedAvg McMahan et al. (2017); Chen et al. (2017), can be $\alpha$-EPR. This is because we can let $\mathcal{D} = \cup_{i\in[m]}\mathcal{W}_i$ and $\alpha$ fraction of the dataset $\mathcal{D}$ can be arbitrarily different from those in $\cup_{i\in S_n}\mathcal{W}_i$. Thus, if a mechanism $c \leq \alpha$, the attack influence is sub-proportional and thus reduced. In Section 6, we show in experiments that $c$-EPR leads to low accuracy in the poisoned datasets.

## 3 LOWER BOUND OF MINIMAX ATTACK LOSS

In this section, we calculate the lower bound of minimax attack loss in both IID and non-IID settings.

## 3.1 LOWER BOUND IN IID SETTING

To show the lower bound on the attack loss that the learned model can achieve, we consider that the server can directly observe the dataset of each worker. Formally, we consider the estimation problem: Nature chooses a ground truth distribution $D$. For each normal worker $i$, nature generates $n$ IID data samples from $D$, $\mathcal{W}_i \sim D^{\otimes n}$. For each poisoned worker $i$, an adversary generates $n$ samples $\mathcal{W}_i$ from a distribution $Q \in \mathcal{P}(\mathcal{Z}^n)$. Upon observing $\{\mathcal{W}_i\}_{i \in [m]}$, the server estimates a hypothesis $\hat{h}(\{\mathcal{W}_i\}_{i=1}^m)$. The estimator can be deterministic or randomized. Let $Mix(\alpha, P, Q) = (1 - \alpha)P + \alpha Q$ be the mixture distribution. When $m \to \infty$, we can consider $\{\mathcal{W}_i\}_{i=1}^m \sim Mix(\alpha, D^{\otimes n}, Q)^{\otimes m}$. Then, the minimax attack loss can be written as:

$$\min_{\hat{h}} \max_D \max_Q \mathbb{E}_{\{\mathcal{W}_i\}_{i=1}^m \sim Mix(\alpha, D^{\otimes n}, Q)^{\otimes m}} R_D(\hat{h}(\{\mathcal{W}_i\}_{i=1}^m)) \quad (4)$$

**Theorem 3.1** (Lower bound in IID setting). *In the IID setting, the lower bound of the minimax attack loss is $\frac{\alpha}{2(1-\alpha)\sqrt{n}}$.*

Therefore, the lower bound for the sample complexity is $n_{\mathcal{H},m}(\epsilon, \delta) \geq \frac{4(1-\alpha)^2}{\alpha^2 \epsilon^2}$.

To prove Theorem 3.1, we can construct two situations that the server cannot distinguish from each other. The ground truth distributions of the two situations are different and any hypothesis leads to at least $\frac{\alpha}{2(1-\alpha)\sqrt{n}}$ attack loss for either of the two situations. The proof will be in Appendix C.1.

## 3.2 LOWER BOUND IN NON-IID SETTING

Then, we consider the lower bound in non-IID setting. We consider a commonly adopted non-IID model in federated learning: Dirichlet distribution Hsu et al. (2019). Let $D_1, \ldots, D_J$ be $J$ data distributions and consider a $J$-dimension vector $\mathbf{p}$ with $\|\mathbf{p}\|_1 = 1$ and a concentration parameter $\gamma$.

The dataset of each worker is sampled as follows: (1) Sample a $J$-dimension vector $\mathbf{q}$ from the Dirichlet distribution $Dir(\gamma\mathbf{p})$. (2) Sample $n$ data samples from mixture probability distribution $Mix'(\mathbf{q}) = \sum_{j=1}^J q_j D_j$. We can see that for each worker $i$, his dataset $\mathcal{W}_i \subseteq \mathcal{Z}^n$ is sampled from a compound probability distribution that $q_i \sim Dir(\gamma\mathbf{p})$ and $\mathcal{W}_i|q_i \sim Mix'(q_i)^{\otimes n}$. We use $\mathcal{C}_\gamma(\mathcal{Z}^n) \subseteq \mathcal{P}(\mathcal{Z}^n)$ to denote the set of such compound probability distributions.

For each concentration parameter $\gamma$, we consider the following estimation problem:

Nature chooses a distribution $C$ from $\mathcal{C}_\gamma(\mathcal{Z}^n)$. Let $C$ be the compound probability distribution and $\mathcal{W} \sim C$ refers to $\mathbf{q} \sim Dir(\gamma\mathbf{p})$ and $\mathcal{W}|\mathbf{q} \sim Mix'(\mathbf{q})^{\otimes n}$. Then, the ground truth distribution induced by $C$ is $C_{gt} = Mix'(\mathbf{p}) = \sum_{j=1}^J p_j D_j$. For each normal worker $i$, nature samples a dataset $\mathcal{W}_i$ from $C$. For each poisoned worker $i$, an adversary generates the $n$-sample dataset $\mathcal{W}_i$ from a distribution $Q \in \mathcal{P}(\mathcal{Z}^n)$. Upon observing $\{\mathcal{W}_i\}_{i=1}^m$, the server estimates a hypothesis $\hat{h}(\{\mathcal{W}_i\}_{i=1}^m)$. Then, the minimax attack loss can be written as:

$$\min_{\hat{h}} \max_{C \in \mathcal{C}_\gamma(\mathcal{Z}^n)} \max_Q \mathbb{E}_{\{\mathcal{W}_i\}_{i=1}^m \sim Mix(\alpha, C, Q)^{\otimes m}} R_{C_{gt}}(\hat{h}(\{\mathcal{W}_i\}_{i=1}^m)) \quad (5)$$

**Theorem 3.2** (Lower bound in non-IID setting). *In the non-IID setting, the lower bound of the minimax attack loss is $\frac{\delta}{2\gamma}$, where $\delta = \max_\delta S_1 \cup S_2$,*

$$S_1 = \left\{ \delta \,\middle|\, \frac{B\left(\frac{\gamma}{2}, \frac{\gamma}{2}\right)}{B\left(\frac{\gamma-\delta}{2}, \frac{\gamma+\delta}{2}\right)} \leq 1 - \frac{\alpha^2}{2(1-\alpha)^2} \right\}, \quad S_2 = \left\{ \delta \,\middle|\, \delta \leq \frac{\gamma\alpha}{\sqrt{n}(1-\alpha)} \right\} \quad (6)$$

*, where $B(\cdot, \cdot)$ is the Beta function. For large $\gamma$ and $n$, the lower bound of the minimax attack loss is $\Omega(\frac{\alpha}{\sqrt{\gamma}} + \frac{\alpha}{\sqrt{n}})$.*

The proof will be in Appendix C.2. Compared with the IID setting, there is an additional attack loss at the scale of $\frac{1}{\sqrt{\gamma}}$. This is because in the non-IID setting, there is less correlation among the datasets of normal workers. Therefore, it is harder for the server to distinguish the poisoned workers from the normal workers.

## 4 ALGORITHM

**Algorithm Overview.** Our algorithm is divided into two phases (shown in Algorithm 1), *i.e.*, the trustworthiness weight update phase and the target model training phase. The first phase returns a weight on workers that represents the trustworthiness of each worker from the server's view. The weight $\mathbf{w} = \{w_i\}_{i=1}^m \in \mathbb{R}^m$ is an $m$-dimensional vector whose elements sum to 1. In the trustworthiness weight update phase, we train a discriminator model and assign a proper weight to each worker. This is the unique and crucial phase of our method. The second phase trains the target model leveraging the weight $\mathbf{w}$.

Let $D_{\mathbf{w}}$ denote the mixture distribution, where the mixing weight is $\mathbf{w}$, and the mixing component is worker datasets. The weight updating phase aims to obtain a weight $\mathbf{w}$ that minimizes $\mathcal{H}$-divergence between $D_{\mathbf{w}}$ and the unknown ground-truth distribution $D$. Then, in the target model training phase, the server trains the target model from the distribution $D_{\mathbf{w}}$ by minimizing the weighted error of the workers, as in Algorithm 1, where $E_i(\theta) = \mathbb{P}_{(x,y) \sim \mathcal{W}_i}[sign \circ g_\theta(x) \neq y]$ is the local error function of worker $i$.

---

**Algorithm 1:** Robust Federated Learning

---

1 // Weight update phase: assign a weight for each worker
2 Let $\mathbf{w} \leftarrow \texttt{WeightUpdate}(\alpha)$;
3 // Training phase: minimize the weighted error
4 $\theta^* \leftarrow \arg\min_{\theta \in \Theta} \sum_{i=1}^m w_i E_i(\theta)$;
5 **return** $sign \circ g_{\theta^*}$;

---

**Trustworthiness weight update phase.** In this phase, we train a discriminator model to differentiate the workers' datasets. The discriminator model is an auxiliary model, different from the target model. The server assigns a proper trustworthiness weight to each worker according to the discriminator model. To train the discriminator model, we first propose the following definition that characterizes the 'variance' of datasets. The following definition can be seen as a variance notion induced by the $\mathcal{H}$-divergence distance notion.

**Definition 4.1** (Variance of datasets)**.** The variance of $m$ datasets $\{\mathcal{W}_i\}_{i=1}^m$ in terms of a weight vector $\mathbf{w}$, a model parameter $\theta$ and a label $\theta$ is

$$Var_{\theta,\mathbf{a}}(\{\mathcal{W}_i\}_{i=1}^m, \mathbf{w}) := \sum_{i=1}^m w_i \left( F_i(\theta, \mathbf{a}) - \sum_{j=1}^m w_j F_j(\theta, \mathbf{a}) \right)^2, \tag{7}$$

$$F_i(\theta, \mathbf{a}) = \sum_{y' \in \mathcal{Y}} a_{y'} \mathbb{P}_{(x,y) \sim \mathcal{W}_i}[y = y', sign(g_\theta(x)) = 1] \tag{8}$$

Then, the weighted variance of $\{\mathcal{W}_i\}_{i=1}^m$ in terms of a hypothesis class $\mathcal{H} = \{sign \circ g_\theta | \theta \in \Theta\}$ is:

$$Var_{\mathcal{H}}(\{\mathcal{W}_i\}_{i=1}^m, \mathbf{w}) := \max_{\theta \in \Theta, \mathbf{a} \in \triangle^1} Var_{\theta,\mathbf{a}}(\{\mathcal{W}_i\}_{i=1}^m, \mathbf{w}) \tag{9}$$

The value $\mathbb{P}_{(x,y) \sim \mathcal{W}_i}[y = y', sign(g_\theta(x)) = 1]$ represents the proportion of input data that originally have a label of $y'$ and are labeled as 1 by the model with parameter $\theta$. Thus, $Var_{\theta,\mathbf{a}}(\{\mathcal{W}_i\}_{i=1}^m, \mathbf{w})$ characterizes the variance of worker datasets in the lens of model parameter $\theta$. When the weight is uniform, we use $Var_{\mathcal{H}}(\{\mathcal{W}_i\}_{i=1}^m)$ to denote the uniformly weighted variance.

We show in Lemma 3 (IID setting) and Lemma 9 (non-IID setting) that the variance over the datasets effectively characterizes the correlation of the datasets. The variance will be small when the datasets are correlated. For the extreme case, if the datasets of workers are duplicated, the variance will be 0.

Since the variance will be small when the datasets are correlated, the variance of the normal workers' dataset is small. Therefore, to bound $\mathcal{H}$-divergence between the weighted distribution $D_{\mathbf{w}}$ and the unknown ground-truth distribution $D$, the weight $\mathbf{w}$ should satisfy: (a) the weighted variance $Var_{\mathcal{H}}(\{\mathcal{W}_i\}_{i=1}^m, \mathbf{w})$ is small; (b) the weight should be as dispersed as possible among the $(1 - \alpha)m$ normal workers. Formally, the weight vector should be in this set:

$$\mathcal{C}_{\beta,\xi,t} = \{\mathbf{w} \in \mathcal{V}_{m,\beta,\xi} : Var_{\mathcal{H}}(\{\mathcal{W}_i\}_{i=1}^m, \mathbf{w}) \leq t\}, \tag{10}$$

$$\mathcal{V}_{m,\beta,\xi} = \left\{ (w_1, \ldots, w_m) : \sum_{i=1}^{m} w_i = 1, \sum_{i \in S_n} w_i = \xi, \text{ and } 0 \le w_i \le \frac{1}{\beta m}, \forall i \right\} \tag{11}$$

, where $\alpha < \beta \le 1 - \alpha$ gives the upper bound of each weight value.

We will prove in that Lemma 5 that if $\mathbf{w} \in \mathcal{C}_{\beta,\xi,t}$, $\mathcal{H}$-divergence between $D_{\mathbf{w}}$ and the unknown ground-truth distribution $D$ can be bounded according to $t$ and $\beta$.

We propose how to find a vector in $\mathcal{C}_{\beta,\xi,t}$ in Algorithm 2. In Algorithm 2, we first initialize a set $S$ to be all workers and let $\mathbf{w}$ be a uniform weight in the set $S$. We also initialize a vector $s \in \mathbb{R}^m$ to be all zeros. Each element $s_i$ in $s$ refers to a score indicating how likely worker $i$ is poisoned. Then, we optimize the parameter $\theta$ to learn a hypothesis that maximizes the variance of workers' datasets (Line 5). Since the variance of normal workers is bounded, the large variance will be mostly contributed by poisoned workers. Therefore, we increase the score $s_i$ for those workers with a larger deviation from the mean (Lines 5- 8). Then, we remove the worker from set $S$ if his score is larger than a threshold $\eta$ (Line 8). We repeat the above steps until $|S|$ is smaller than $(1 - 2\alpha)$.

---

**Algorithm 2:** WeightUpdate

---

1 **Function** WeightUpdate($\alpha$)**:**
2   // Initialization
3   $S \leftarrow [m]$, $\mathbf{w} \leftarrow \{\frac{1}{m}, \ldots, \frac{1}{m}\}$, $s \leftarrow \{0, \ldots, 0\}$;
4   **for** $r = 1, 2, \ldots$ **do**
5     Let $\theta, \mathbf{a} \leftarrow \arg\max_{\theta, \mathbf{a}} \sum_{i=1}^{m} w_i \left( F_i(\theta, \mathbf{a}) - \sum_{j=1}^{m} w_j F_j(\theta, \mathbf{a}) \right)^2$;
6     For $i \in S$, $\tau_i \leftarrow \left( F_i(\theta, \mathbf{a}) - \sum_{j=1}^{m} w_j F_j(\theta, \mathbf{a}) \right)^2$;
7     $\tau_{max} \leftarrow \max_{i \in S} \tau_i$;
8     For $i \in S$, $s_i \leftarrow s_i + \frac{\tau_i}{\tau_{max}}$. If $s_i \ge \eta$, then remove $i$ from $S$;
9     **if** $|S| < (1 - 2\alpha)m$ **then**
10       **return w**;
11     **end**
12     **else**
13       Update the weight $\mathbf{w}$ such that $w_i \leftarrow \frac{1}{|S|}$ for $i \in S$ and $w_i \leftarrow 0$ otherwise;
14     **end**
15   **end**

---

We prove in Lemma C.5 that our weight update phase can find weight in $\mathcal{C}_{\beta,\xi,t}$ and prove in Theorem 5.1 (IID setting) and Theorem A.1 (non-IID setting) that our algorithm asymptotically matches the lower bound in both IID and non-IID settings.

To realize the maximization step in the distributed setting, we first replace the $sign$ function by the sigmoid function. Then, it can be implemented by generalized FedAVG, where the workers send the gradients and the model output to the server, described in Appendix F. The pseudo-code of the variance maximization process is in Algorithm 4 in Section D.1.

**Target model training phase.** In the training phase, the system trains the target model to minimize the weighted error. The phase can be implemented directly by FedAVG. The server collects the gradients from workers and sums them with $\mathbf{w}$ as the weight, where $\mathbf{w}$ is obtained in the trustworthiness weight update phase.

## 5 ANALYSIS

We first analyze our algorithm in the IID setting in this section. We show that our algorithm achieves $c$-EPR, where $c = \tilde{O}((\frac{1}{n})^{\frac{1}{2}} + (\frac{d}{mn})^{\frac{1}{2}})$. In addition, the error of the learning task can also bounded by $\tilde{O}((\frac{1}{n})^{\frac{1}{2}} + (\frac{d}{mn})^{\frac{1}{2}})$. We state the EPR and the attack loss bound in Theorem 5.1. Theorem 5.1 is proved in Appendix, where we break down the proof into Lemma 3- 6.

**Theorem 5.1** (Main result in IID setting). *When $\alpha \le \frac{1}{3}$, $n > 1 + \frac{d}{m}$, there exists a constant $C'$ and $C = (\sqrt{\frac{\alpha}{1-3\alpha}}\frac{4-9\alpha}{1-3\alpha} + \sqrt{\frac{2\alpha}{1-3\alpha}})C'$. Then, with probability $1-\delta$, Algorithm 1 satisfies:*

- $C(\sqrt{\frac{1}{n}} + \sqrt{\frac{d\ln(mn/d)+\ln(1/\delta)}{mn}})$-*EPR*

- *The attack loss of our algorithm is* $C(\sqrt{\frac{1}{n}} + \sqrt{\frac{d\ln(mn/d)+\ln(1/\delta)}{mn}})$

For the non-IID setting, we show that our algorithm achieves an attack loss roughly $\sqrt{\frac{1}{\gamma}}$ more than that in the IID setting in Appendix A.

## 6 EXPERIMENT

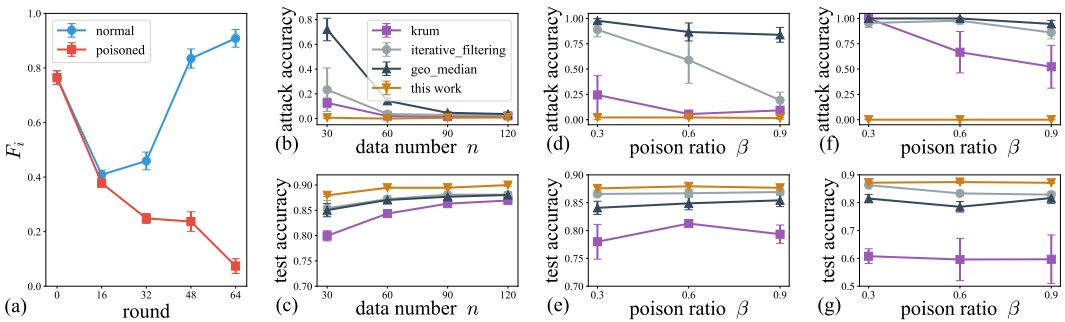

Figure 2: Experimental results. (b-e) are conducted in IID setting. (f,g) are conducted in non-IID setting. Figures in the top row (b,d,f) show the attack accuracy. Figures in the bottom row (c,e,g) show the test accuracy. We adopt standard error as the error bar.

We evaluate robust federated learning mechanisms in MNIST and CIFAR-10 datasets with the flip-label attack and backdoor attack. In this section, we focus on the result of defending the flip-label attack in MNIST dataset. More details and experiments are provided in Appendix D.

**Training process of the discriminator model.** We first show an example of the discriminator model training process to show that our discriminator model is effective in differentiating the unpoisoned datasets and poisoned datasets with a large number of workers. In Figure(a), we consider 2000 normal workers and 500 workers with poisoned datasets. Each normal worker dataset consists of 30 IID data samples. Each poisoned dataset consists of 30 label-flipped data samples. We see from Figure(a) that as the discriminator model maximizes the dataset variance, the unpoisoned datasets and poisoned datasets are differentiated.

### 6.1 COMPARING WITH BASELINES

**Baselines.** We consider the commonly adopted and theoretically guaranteed poisoning attack defenses: geometric median method Blanchard et al. (2017); Chen et al. (2017), iterative filtering method Su & Xu (2018), and Krum function method Shejwalkar et al. (2022). We set the learning rate and the epoch number of the baselines and our training phase to be the same, where the learning rate $\zeta = 0.01$ and the training epoch is 20. The server selects 20% of workers at each round.

**Metric.** To characterize the attack loss, we use the accuracy of the trained target model on the MNIST test dataset. The larger the test accuracy, the smaller the attack loss will be. To characterize the adversary's gain, we use the accuracy of the trained target model on the poisoned data samples, denoted as attack accuracy. Here, the poisoned data samples are the flipped label data samples that the adversary put in the corrupted workers' datasets.

**Result** We consider 200 normal workers, 50 workers with poisoned data, and $\eta = 1$ in Algorithm 2 Line 8. Figure 2(b,c) demonstrates how the effectiveness of the defenses varies with the number of

IID data samples $n$ each worker has. With larger $n$, the datasets of normal workers converge. Thus, it will be easier to defend against the adversary. Our algorithm maintains higher test accuracy and lower attack accuracy than the baselines, especially when $n$ is small.

In Figure 2(d-g), the adversary camouflages the poisoned data samples with the normal workers' data samples. We use the poison rate $\beta$ to denote the ratio of poisoned data in each dataset. We consider 30 samples for each worker. In Figure 2(d,e), we consider the IID setting. In Figure 2(f,g), we consider the non-IID setting. The normal workers' data are non-IID over the different labels Hsu et al. (2019), and the concentration parameter is $1.0$. Our algorithm maintains higher test accuracy and lower attack accuracy than the baselines in both IID and non-IID settings.

In addition, our algorithm is more efficient compared to the baselines. The commutation complexity of both phases in our algorithm is the same as that of FedAVG. In comparison,in the baseline algorithms, the server conducts complicated gradient aggregation operations. When $m$ is large, this will be the primary cause of the algorithm's slowdown.

## 7 Related work

**Robust gradient aggregation:** Robust gradient aggregation is initially a defense to the model poisoning attack is gradient robust aggregation Chen et al. (2017); Su & Xu (2018); Yin et al. (2018); Blanchard et al. (2017). The method can be applied to defend against the data poisoning attack. However, these algorithms are not effective enough because their performance depends on the variance of gradients calculated by the workers, which are vectors with high dimensions and grow as the models become larger. In federated learning with robust gradient aggregation, the server uses robust mean aggregation to aggregate those gradients, instead of averaging the gradients collected from the workers as in FedAVG McMahan et al. (2017). Different robust aggregation techniques provide bounds of the resulting model's attack loss based on different norms of the covariance matrix of gradients $\Sigma_g$ and the number of model parameters (dimension of gradients) $d_g$. For example, if dimension-wise median Yin et al. (2018), or geometric median Feng et al. (2014); Chen et al. (2017) are used as the robust aggregation technique, the attack loss can be bounded by $O((\frac{trace(\Sigma_g)}{n})^{\frac{1}{2}} + (\frac{d_g}{mn})^{\frac{1}{2}})$ Feng et al. (2014); Yin et al. (2018). For iterative filtering Su & Xu (2018); Steinhardt et al. (2017); Diakonikolas et al. (2019) method, the attack loss can be bounded by $O((\frac{\|\Sigma_g\|_2}{n})^{\frac{1}{2}} + (\frac{d_g}{mn})^{\frac{1}{2}})$ Su & Xu (2018), where $\|\Sigma_g\|_2$ is $L2$-norm of the covariance matrix. For Krum function Blanchard et al. (2017), the number of samples $n$ should be comparable with the number of model parameters $d_g$ for their theoretical bound. With the increasing number of parameters of the model, the above attack loss bounds expand, even when $m$ goes to infinity. For the trace of the covariance matrix, it explicitly depends on the dimension of the gradients. For the $L2$-norm of the covariance matrix, it can also go to infinity as the width of the neuron network model goes to infinity Karakida et al. (2021).

**Data poisoning attack in federated learning:** Tolpegin et al. Tolpegin et al. (2020) adopt principal component analysis (PCA) to detect the outlier gradients as their defense against the data poisoning attack. PCA over gradients is also one of the robust gradient aggregation methods. Kairouz et al. Kairouz et al. (2021) and Han et al. Han et al. (2024) consider both data poisoning attack and model poisoning attack. They also adopt robust gradient aggregation methods as their defenses against the data poisoning attack.

**Random perturbation:** Naseri et al. Naseri et al. (2020) explore the similarity between protecting privacy and robustness. Random perturbation is more effective when the number of poisoned data is smaller. Since our EPR property limits the number of poisoned data, random perturbation and our mechanism can complement each other.

## 8 Conclusion

We propose a mechanism that asymptotically achieves the lower bound of attack loss in both IID setting and non-IID setting, when $\alpha < \frac{1}{3}$. In addition, we propose a metric EPR for data poisoning attacks in the federated learning system to characterize the effectiveness of the adversary. Our mechanism also achieves a low EPR.

## 9    ACKNOWLEDGEMENT

This work is supported by Tsinghua University Dushi Program and Shanghai Qi Zhi Institute Innovation Program.

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

## A  ANALYSIS IN NON-IID SETTING

In the non-IID setting, the main difference is that the dataset variance of the normal worker is $\frac{1}{2(\gamma+1)}$ larger than that in the IID-setting. Therefore, the EPR and attack loss are as follows:

**Theorem A.1** (Main result in non-IID setting). *Let the dataset of each normal worker be sampled from a distribution $C \in \mathcal{C}_\gamma(\mathcal{Z}^n)$. When $\alpha \leq \frac{1}{3}$, $n > 1 + \frac{d}{m}$, there exists a constant $C'$ and $C = (\sqrt{\frac{\alpha}{1-2\alpha}}\frac{4-9\alpha}{1-3\alpha} + \frac{\sqrt{\alpha(2-3\alpha)}}{1-2\alpha})C'$, $E = C(\sqrt{\frac{1}{\gamma}} + \sqrt{\frac{1}{n}} + \sqrt{\frac{d\ln(mn/d)+\ln(1/\delta)}{mn}} + \sqrt[4]{\frac{\ln(1/\delta)}{2m}})$. Then, with probability $1 - \delta$, Algorithm 1 satisfies:*

- *E-EPR*

- *The attack loss of our algorithm is also $E$*

We can see the upper bound asymptotically matches the lower bound when $m \to \infty$.

## B  DETAIL OF FEDAVG

FedAVG McMahan et al. (2017); Chen et al. (2017) is a basic federated learning mechanism in which the worker calculates the gradient of his local dataset and the server collects the gradients and updates the model with the weighted average of the gradients. The mechanism is described in Algorithm 3, where $\zeta_r$ is the learning rate of round $r$.

---

**Algorithm 3:** FedAVG

1 **Worker $i$:**
2 If selected in $S$ at round $r$, compute $grad_{i,r} \leftarrow \nabla F_i(\theta_r)$;
3 Send $grad_{i,r}$ to server;
4 **Server:**
5 **for** $r = 1, 2, \ldots$ **do**
6     $S \leftarrow$ (random set of $s$ workers by weight **w**);
7     Receive $grad_{i,r}$ from worker $i \in S$;
8     $\theta_{r+1} \leftarrow \theta_r - \zeta_r \sum_{i \in S} grad_{i,r}$;
9 **end**

---

## C  PROOFS

### C.1  PROOF OF THEOREM 3.1

*Proof.* Let $D(h, \Delta, +)$ be a data distribution that $\mathbb{P}((x, y)) = \frac{1}{|\mathcal{X}|}(\frac{1}{2} + h(x) * y * \Delta)$. Let $D(h, \Delta, -)$ be a data distribution that $\mathbb{P}((x, y)) = \frac{1}{|\mathcal{X}|}(\frac{1}{2} - h(x) * y * \Delta)$.

Let $H^2(P, Q) = \mathbb{E}_Q\left[\left(1 - \sqrt{\frac{P}{Q}}\right)^2\right]$ be the squared Hellinger distance. Then,

$$H^2\left(D(h, \Delta, +), D(h, \Delta, -)\right) = 2 - 2\sqrt{1 - 4\Delta^2} \tag{12}$$

and

$$\begin{aligned}
&H^2\left(D(h, \Delta, +)^{\otimes n}, D(h, \Delta, -)^{\otimes n}\right) \\
=&2 - 2\left(1 - \frac{1}{2}H^2\left(D(h, \Delta, +), D(h, \Delta, -)\right)\right)^n \\
=&2 - 2(1 - 4\Delta^2)^{n/2} \\
\leq&4n\Delta^2
\end{aligned} \tag{13}$$

The total variant distance between $D(h, \Delta, +)^{\otimes n}$ and $D(h, \Delta, -)^{\otimes n}$ can be bound as:

$$d_{\mathrm{TV}}(D(h, \Delta, +)^{\otimes n}, D(h, \Delta, -)^{\otimes n})$$
$$\leq H(D(h, \Delta, +)^{\otimes n}, D(h, \Delta, -)^{\otimes n}) \leq 2\sqrt{n}\Delta \quad (14)$$

Let $\Delta = \frac{\alpha}{2(1-\alpha)\sqrt{n}}$. Then,

$$d_{\mathrm{TV}}\left(D(h, \Delta, +)^{\otimes n}, D(h, \Delta, -)^{\otimes n}\right) \leq \epsilon/(1-\epsilon) \quad (15)$$

By Lemma 1, there exists distributions $Q_+$ and $Q_-$ that $(1-\alpha)D(h, \Delta, +)^{\otimes n} + \alpha Q_+ = (1-\alpha)D(h, \Delta, -)^{\otimes n} + \alpha Q_-$. Thus, for any hypothesis, the server will estimate the hypothesis $\hat{h}$ at the same probability whether the data sets are sampled from $(1-\alpha)D(h, \Delta, +)^{\otimes n} + \alpha Q_+$ or $(1-\alpha)D(h, \Delta, -)^{\otimes n} + \alpha Q_-$.

For any $\hat{h}$,

$$\epsilon_{D(h, \Delta, +)}(\hat{h}) + \epsilon_{D(h, \Delta, -)}(\hat{h})$$
$$= \mathbb{P}_{(x,y) \sim D(h, \Delta, +)}[\hat{h}(x) \neq y] + \mathbb{P}_{(x,y) \sim D(h, \Delta, -)}[\hat{h}(x) \neq y] \quad (16)$$
$$= 1$$

Therefore, $R_{D(h, \Delta, +)}(\hat{h}) + R_{D(h, \Delta, -)}(\hat{h}) = 1 - 2*(\frac{1}{2} - \Delta) = 2\Delta$.

Thus,

$$\min_{\hat{h}} \max_{D} \max_{Q} \mathbb{E}_{\{\mathcal{W}_i\}_{i=1}^m \sim Mix(\alpha, D, Q)^{\otimes m}} R_D(\hat{h}(\{\mathcal{W}_i\}_{i=1}^m))$$
$$\geq \frac{1}{2}(R_{D(h, \Delta, +)}(\hat{h}) + R_{D(h, \Delta, -)}(\hat{h})) \quad (17)$$
$$= \frac{\alpha}{2(1-\alpha)\sqrt{n}}$$

$\square$

**Lemma 1** (Chen et al. (2015)). *Consider two distributions $P_1$ and $P_2$ such that $d_{TV}(P_1, P_2) \leq \frac{\epsilon}{1-\epsilon}$. Then, there exists distributions $Q_1$ and $Q_2$ that $(1-\epsilon')P_1 + \epsilon'Q_1 = (1-\epsilon')P_2 + \epsilon'Q_2$.*

### C.2    PROOF OF THEOREM 3.2

*Proof.* Consider a concentration parameter $\gamma$ and a Dirichlet distribution $Dir(\gamma_1, \gamma_2)$, where $\gamma_1 + \gamma_2 = \gamma$. Let $\delta = \gamma_2 - \gamma_1$. Then, $\gamma_1 = \frac{\gamma - \delta}{2}$, $\gamma_2 = \frac{\gamma + \delta}{2}$. Let $f_{Dir}(\cdot; \gamma_1, \gamma_2)$ be the corresponding probability density function. Let $D(h, +)$ be a data distribution that $\mathbb{P}((x, y)) = \frac{1}{|\mathcal{X}|}(\frac{1 + h(x)*y}{2})$. Let $D(h, -)$ be a data distribution that $\mathbb{P}((x, y)) = \frac{1}{|\mathcal{X}|}(\frac{1 - h(x)*y}{2})$.

Let $\mathcal{S}_1, \mathcal{S}_2 \in \mathcal{C}_\gamma(\mathcal{Z}^n)$ be two distributions of workers' datasets defined in Section 3.2.

- For $\mathcal{S}_1$, the parameters are $\mathbf{p} = (\frac{\gamma_1}{\gamma}, \frac{\gamma_2}{\gamma})$, $D_1 = D(h, -)$, and $D_2 = D(h, +)$. Let $D_{gt,1}$ denote the corresponding ground-truth distribution. Let $f_1(\cdot; \mathbf{q})$ be the probability density function of the $n$-fold mixture distribution $(q_1 D(h, -) + q_2 D(h, +))^{\otimes n}$.

- For $\mathcal{S}_2$, the parameters are $\mathbf{p} = (\frac{\gamma_1}{\gamma}, \frac{\gamma_2}{\gamma})$, $D_1 = D(h, +)$, and $D_2 = D(h, -)$. Let $D_{gt,2}$ denote the corresponding ground-truth distribution. Let $f_2(\cdot; \mathbf{q})$ be the probability density function of the $n$-fold mixture distribution $(q_1 D(h, +) + q_2 D(h, -))^{\otimes n}$.

Then,

$$d_{TV}(\mathcal{S}_1, \mathcal{S}_2) = \int |\mathcal{S}_1 - \mathcal{S}_2|$$

$$= \int_{\mathcal{W} \in \mathcal{Z}^n} \left| \int_{\|\mathbf{q}\|_1 = 1} f_1(\mathcal{W}; \mathbf{q}) f_{Dir}(\mathbf{q}; \gamma_1, \gamma_2) d\mathbf{q} - \int_{\|\mathbf{q}\|_1 = 1} f_2(\mathcal{W}; \mathbf{q}) f_{Dir}(\mathbf{q}; \gamma_1, \gamma_2) d\mathbf{q} \right|$$

$$= \int_{\mathcal{W} \in \mathcal{Z}^n} \left| \int_{\|\mathbf{q}\|_1 = 1} f_1(\mathcal{W}; \mathbf{q}) (f_{Dir}(\mathbf{q}; \gamma_1, \gamma_2) - f_{Dir}(\mathbf{q}; \gamma_2, \gamma_1)) d\mathbf{q} \right| \qquad (18)$$

$$\leq \int_{\|\mathbf{q}\|_1 = 1} |f_{Dir}(\mathbf{q}; \gamma_1, \gamma_2) - f_{Dir}(\mathbf{q}; \gamma_2, \gamma_1)| \, d\mathbf{q}$$

$$\leq \sqrt{2 - 2 \frac{B(\frac{\gamma_1 + \gamma_2}{2}, \frac{\gamma_1 + \gamma_2}{2})}{B(\gamma_2, \gamma_1)}}$$

The last inequality follows from the below Lemma 2. In addition,

$$d_{TV}(\mathcal{S}_1, \mathcal{S}_2) = \int |\mathcal{S}_1 - \mathcal{S}_2|$$

$$= \int_{\mathcal{W} \in \mathcal{Z}^n} \left| \int_{\|\mathbf{q}\|_1 = 1} f_1(\mathcal{W}; \mathbf{q}) f_{Dir}(\mathbf{q}; \gamma_1, \gamma_2) d\mathbf{q} - \int_{\|\mathbf{q}\|_1 = 1} f_2(\mathcal{W}; \mathbf{q}) f_{Dir}(\mathbf{q}; \gamma_1, \gamma_2) d\mathbf{q} \right|$$

$$\leq \int_{\mathcal{W} \in \mathcal{Z}^n} \int_{\|\mathbf{q}\|_1 = 1} |f_1(\mathcal{W}; \mathbf{q}) - f_2(\mathcal{W}; \mathbf{q})| \, f_{Dir}(\mathbf{q}; \gamma_1, \gamma_2) \qquad (19)$$

$$= \int_{\mathcal{W} \in \mathcal{Z}^n} \int_{\|\mathbf{q}\|_1 = 1} |f_1(\mathcal{W}; \mathbf{q}) - f_1(\mathcal{W}; 1 - \mathbf{q})| \, f_{Dir}(\mathbf{q}; \gamma_1, \gamma_2)$$

$$\leq \int_{\|\mathbf{q}\|_1 = 1} 2\sqrt{n} (\frac{1}{2} - q_1) f_{Dir}(\mathbf{q}; \gamma_1, \gamma_2) = \sqrt{n} (\frac{\gamma_2 - \gamma_1}{\gamma})$$

Let set $S_1 = \{\delta \mid \frac{B(\frac{\gamma}{2}, \frac{\gamma}{2})}{B(\frac{\gamma - \delta}{2}, \frac{\gamma + \delta}{2})} \leq 1 - \frac{\alpha^2}{2(1-\alpha)^2}\}$, $S_2 = \{\delta \mid \delta \leq \frac{\gamma\alpha}{\sqrt{n}(1-\alpha)}\}$, and $\delta = \max_\delta S_1 \cup S_2$. Then,

$$\min_{\hat{h}} \max_{C \in \mathcal{C}_\gamma(\mathcal{Z}^n)} \max_Q \mathbb{E}_{\{\mathcal{W}_i\}_{i=1}^m \sim Mix(\alpha, C, Q)^{\otimes m}} R_{C_{gt}}(\hat{h}(\{\mathcal{W}_i\}_{i=1}^m))$$

$$\geq \frac{1}{2}(R_{D_{gt,1}}(\hat{h}) + R_{D_{gt,2}}(\hat{h})) \qquad (20)$$

$$= \frac{\delta}{2\gamma}$$

To estimate the asymptotic behavior, we can approximate gamma functions by Stirling's approximation. Let $w = \frac{\delta}{\sqrt{\gamma}}$,

$$\ln \frac{B(\frac{\gamma}{2}, \frac{\gamma}{2})}{B(\frac{\gamma - w\sqrt{\gamma}}{2}, \frac{\gamma + w\sqrt{\gamma}}{2})}$$

$$= (\gamma - 1) \ln \frac{\gamma}{2} - \left( \frac{\gamma - w\sqrt{\gamma} - 1}{2} \right) \ln \frac{\gamma - w\sqrt{\gamma}}{2}$$

$$- \left( \frac{\gamma + w\sqrt{\gamma} - 1}{2} \right) \ln \frac{\gamma + w\sqrt{\gamma}}{2} + O\left( \gamma^{-2} \right)$$

$$= - \frac{\gamma - w\sqrt{\gamma} - 1}{2} \ln(1 - \frac{w^2}{\gamma})$$

$$- w\sqrt{\gamma} \ln(1 + \frac{w\sqrt{\gamma}}{\gamma}) + O\left( \gamma^{-2} \right)$$

$$= - \frac{w^2}{2} + O(\gamma^{-\frac{1}{2}})$$

Therefore,

$$
\min_{\hat{h}} \max_{C \in \mathcal{C}_\gamma(\mathcal{Z}^n)} \max_Q \mathbb{E}_{\{\mathcal{W}_i\}_{i=1}^m \sim Mix(\alpha, C, Q)^{\otimes m}} R_{C_{gt}}(\hat{h}(\{\mathcal{W}_i\}_{i=1}^m))
$$

$$
\geq \max_\delta S_1 \cup S_2
$$

$$
\geq \max\{\sqrt{-\frac{1}{2\gamma}\log(1 - \frac{\alpha^2}{2(1-\alpha)^2})} + O(\gamma^{-1}), \frac{\alpha}{2(1-\alpha)\sqrt{n}}\} \tag{21}
$$

$$
= \Omega(\frac{\alpha}{\sqrt{\gamma}} + \frac{\alpha}{\sqrt{n}})
$$

$\square$

**Lemma 2.** $d_{TV}(Dir(\gamma_1, \gamma_2), Dir(\gamma_2, \gamma_1)) \leq \sqrt{2 - 2\frac{B(\frac{\gamma_1+\gamma_2}{2}, \frac{\gamma_1+\gamma_2}{2})}{B(\gamma_2, \gamma_1)}}$

*Proof.* Let $H^2(P, Q)$ be the squared Hellinger distance.

$$
H^2\left(Dir(\gamma_1, \gamma_2), Dir(\gamma_2, \gamma_1)\right)
$$

$$
= 2 - 2\int \sqrt{f(x; \gamma_1, \gamma_2)f(x; \gamma_2, \gamma_1)}
$$

$$
= 2 - 2\int \frac{1}{B(\gamma_2, \gamma_1)} x^{\frac{\gamma_1+\gamma_2}{2}-1}(1-x)^{\frac{\gamma_1+\gamma_2}{2}-1} \tag{22}
$$

$$
= 2 - 2\frac{B(\frac{\gamma_1+\gamma_2}{2}, \frac{\gamma_1+\gamma_2}{2})}{B(\gamma_2, \gamma_1)}
$$

By the relation between total variant distance and the squared Hellinger distance,

$$
d_{TV}(Dir(\gamma_1, \gamma_2), Dir(\gamma_2, \gamma_1)) \leq \sqrt{2 - 2\frac{B(\frac{\gamma_1+\gamma_2}{2}, \frac{\gamma_1+\gamma_2}{2})}{B(\gamma_2, \gamma_1)}} \tag{23}
$$

$\square$

### C.3 PROOF OF THEOREM 5.1

We summarize the proof of Theorem 5.1 in the following lemmas, and the proofs of the Lemmas are in Appendix C.4- C.8. We first prove that the variance of normal workers is bounded in Lemma 3.

**Lemma 3** (Statistic of IID normal workers). *Let $\{\mathcal{W}_i\}_{i \in S_n}$ be $|S_n|$ datasets that each dataset consists of $n$ data sampled independently from distribution $D$. When $n > 1 + \frac{d}{m}$, there exists a constant $C$ that with probability $1 - \delta$,*

$$
Var_{\mathcal{H}}(\{\mathcal{W}_i\}_{i \in S_n}) \leq C\left(\frac{1}{n} + \frac{d\ln(mn/d) + \ln(1/\delta)}{mn}\right) \tag{24}
$$

In Lemma 4, we prove that when the variance of normal workers is bounded, Algorithm 2 can output a weight such that the weighted variance of all workers is also bounded by a similar bound.

**Lemma 4** (Property of $WeightUpdate$). *For $\alpha < \frac{1}{3}$, let $\mathbf{w}$ be the weight vector output by $WeightUpdate(\alpha)$ in Algorithm 2. Then, $\mathbf{w} \in C_{\beta, \xi, t}$, where $\xi \geq \frac{1-3\alpha}{1-2\alpha}$, $\beta \geq 1 - 2\alpha$, $t < \frac{4-9\alpha}{1-3\alpha}Var_{\mathcal{H}}(\{\mathcal{W}_i\}_{i \in S_n})$.*

Since the above Lemma 4 proves that Algorithm 2 output a weight vector in $\mathcal{C}_{\beta, \xi, t}$, in Lemma 5, we prove that the distance between the weighted distribution $D_{\mathbf{w}}$ and the unknown distribution $D$ can be bounded by $t$ and $\beta$ when $\mathbf{w} \in \mathcal{C}_{\beta, \xi, t}$.

**Lemma 5** (Property of $\mathcal{C}_{\beta, \xi, t}$). *For $\mathbf{w} \in \mathcal{C}_{\beta, \xi, t}$,*

$$
d_{\mathcal{H}}(D, D_{\mathbf{w}}) \leq \sqrt{\frac{1-\xi}{\xi}t} + \sqrt{\frac{1-\alpha-\beta\xi}{\beta\xi}Var_{\mathcal{H}}(\{\mathcal{W}_i\}_{i \in S_n}} \\
+ d_{\mathcal{H}}(D, \cup_{i \in S_n}\mathcal{W}_i) \tag{25}
$$

Since $d_{\mathcal{H}}(D, \{\mathcal{W}_i\}_{i \in S_n})$ is the $\mathcal{H}$-divergence between $D$ and $|S_n|n$ samples independently sampled from $D$. By Lemma 1 in Ben-David et al. (2010), $d_{\mathcal{H}}(D, \{\mathcal{W}_i\}_{i \in S_n}) = \tilde{O}((\frac{d}{nm})^{\frac{1}{2}})$. Therefore, $d_{\mathcal{H}}(D, D_{\mathbf{w}}) = \tilde{O}((\frac{1}{n})^{\frac{1}{2}} + (\frac{d}{nm})^{\frac{1}{2}})$. Then, the EPR bound in Theorem 5.1 directly follows the above lemmas.

For the attack loss, the following lemma shows the relation between $d_{\mathcal{H}}(D, D_{\mathbf{w}})$ and $|\epsilon_D(h) - \epsilon_{D_{\mathbf{w}}}(h)|$.

**Lemma 6.** *Let $h$ be a hypothesis in class $\mathcal{H}$. Then, $|\epsilon_D(h) - \epsilon_{D'}(h)| \leq 2d_{\mathcal{H}}(D, D')$.*

### C.4   PROOF OF LEMMA 3

*Proof.* Let $F_{i,y'}(\theta) = \mathbb{P}_{(x,y) \sim \mathcal{W}_i}[y = y', sign(g_\theta(x)) = 1]$. Let $m' = |S_n|$. Draw $m'n$ samples from the dataset and randomly partition them into $m'$ groups of size $n$. Define $s = \lceil i/n \rceil$ and $r = i - (s-1)n$, and let $X_i$ be the sample assigned to position $r$ in worker $s$'s local dataset. Consider an $m'$-dimensional vector $\mathbf{v}$ with entries $v_i$. Define $\bar{F}_{i,y}(\theta) = F_{i,y}(\theta) - \mathbb{E}F_{i,y}(\theta)$ and $\bar{F}_{\mathbf{v},y}(\theta) = \sum_{i=1}^{m'} v_i \bar{F}_{i,y}(\theta)$. We construct the Doob martingale

$$Z_{\mathbf{v},k,y,\theta} = \mathbb{E}[\bar{F}_{\mathbf{v},y}(\theta) \mid X_1, \ldots, X_k]. \tag{26}$$

Since $|Z_{\mathbf{v},k,y,\theta} - Z_{\mathbf{v},k-1,y,\theta}| \leq \frac{\|\mathbf{v}\|}{n\sqrt{m'}}$, the Azuma–Hoeffding inequality yields

$$\mathbb{E}[e^{Z_{\mathbf{v},m'n,y,\theta}}] \leq \exp\left(\frac{\|\mathbf{v}\|^2}{8n}\right). \tag{27}$$

Let $Z_{i,m'n,y,\theta} = Z_{\mathbf{e}_i,m'n,y,\theta}$, where $\mathbf{e}_i$ is the $i$-th standard basis vector. Then, for any $t < 2n$,

$$\mathbb{E}\left[\exp\left(t \sum_{i=1}^{m'} Z_{i,m'n,y,\theta}^2\right)\right] \leq \left(1 - \frac{t}{2n}\right)^{-m'/2}. \tag{28}$$

Applying Chernoff's bounding method,

$$\mathbb{P}\left[\sum_{i=1}^{m'} Z_{i,m'n,y,\theta}^2 > \epsilon\right] \leq e^{-t\epsilon}(1 - \frac{t}{2n})^{-m'/2} \tag{29}$$

The above probability is calculated given the $\theta$. Since $d$ is the VC-dimension of the hypothesis class induced by $\Theta$, by the uniform bound, for any $\theta \in \Theta$,

$$\mathbb{P}[\sum_{i=1}^{m'} Z_{i,m'n,y,\theta}^2 \geq \frac{m'}{4n} + \epsilon, \forall \theta] \leq (\frac{em'n}{d})^d e^{-t\epsilon} e^{-\frac{tm'}{4n}}(1 - \frac{t}{2n})^{-m'/2} \tag{30}$$

Therefore, with probability $1 - \delta$,

$$\begin{aligned}
&Var_{\mathcal{H}}(\{\mathcal{W}_i\}_{i \in s}) \\
&\leq \frac{4}{n} \\
&\quad + \min_t \frac{\ln(1/\delta) + d\ln(em'n/d) - tm'/4n - (m'/2)\ln(1 - t/2n)}{m't} \\
&= O\left(\frac{1}{n} + \frac{\sqrt{d\ln(mn/d) + \ln(1/\delta)}}{\sqrt{mn}}\right) \\
&= O\left(\frac{1}{n} + \frac{d\ln(mn/d) + \ln(1/\delta)}{mn}\right)
\end{aligned} \tag{31}$$

$\square$

## C.5 PROOF OF LEMMA 4

*Proof.* Since $|S| < (1 - 2\alpha)m$, it is evident from the algorithm that $\xi \geq \frac{1-3\alpha}{1-2\alpha}$ and $\beta \geq 1 - 2\alpha$.

In the following, we show that if the algorithm outputs,

$$k := \frac{Var_{\mathcal{H}}(\{\mathcal{W}_i\}_{i=1}^m, \mathbf{w})}{Var_{\mathcal{H}}(\{\mathcal{W}_i\}_{i \in S_n})} \leq \frac{4 - 9\alpha}{1 - 3\alpha}.$$

We prove this by contradiction. Assume that $k \geq \frac{4-9\alpha}{1-3\alpha}$. We show that in this case, the algorithm will not output.

We define some notations. For $\mathbf{v} = \{v_i\}_{i=1}^m$ and $\mathbf{w} = \{w_i\}_{i=1}^m$, define the weighted mean and variance be

$$\mu(\mathbf{v}, \mathbf{w}) = \frac{\sum_{i=1}^m w_i v_i}{\sum_{i=1}^m w_i} \quad \text{and} \quad Var(\mathbf{v}, \mathbf{w}) = \frac{1}{\sum_{i=1}^m w_i} \sum_{i=1}^m w_i (v_i - \mu(\mathbf{v}, \mathbf{w}))^2.$$

We partition $\mathbf{w} = \mathbf{w}^n + \mathbf{w}^p$ as follows: $\mathbf{w}^n$ is the $m$-dimensional vector with

$$\mathbf{w}_i^n = \begin{cases} \mathbf{w}_i & \text{if } i \in \mathcal{S}_n, \\ 0 & \text{if } i \in \mathcal{S}_p, \end{cases}$$

and $\mathbf{w}^p$ is the $m$-dimensional vector with

$$\mathbf{w}_i^p = \begin{cases} \mathbf{w}_i & \text{if } i \in \mathcal{S}_p, \\ 0 & \text{if } i \in \mathcal{S}_n. \end{cases}$$

Then, we define

$$\alpha_{n,r} := \frac{\|\mathcal{S}_n \cap \mathcal{S}_r\|}{m}, \quad \alpha_{p,r} := \frac{\|\mathcal{S}_p \cap \mathcal{S}_r\|}{m}.$$

Let $\tau$ be the $m$-dimensional vector whose $i$-th element is $\tau_i$ defined in Section 4.

**Inductive claim.** We prove by induction that at each round $r$, when $k \geq \frac{4-9\alpha}{1-3\alpha}, \eta \geq \sqrt{\frac{3\alpha}{1-3\alpha}} + 1$,

$$\frac{\alpha_{n,r}}{\alpha_{p,r}} \geq \frac{1 - 2\alpha + \frac{\alpha}{1+\eta}}{\alpha},$$

and

$$\alpha_{n,r} \geq 1 - 2\alpha.$$

Thus, $|S| \geq (1 - 2\alpha)m$.

**Inductive step.** By the constraint $k Var(\tau, \mathbf{w}^n) = Var(\tau, \mathbf{w})$ and the variance decomposition formula, we obtain

$$k(\alpha_{n,r} + \alpha_{p,r})Var(\tau, \mathbf{w}^n) = \alpha_{n,r} Var(\tau, \mathbf{w}^n) + \alpha_{p,r} Var(\tau, \mathbf{w}^p)$$
$$+ \alpha_{n,r}(\mu(\tau, \mathbf{w}^n) - \mu(\tau, \mathbf{w}))^2 \tag{32}$$
$$+ \alpha_{p,r}(\mu(\tau, \mathbf{w}^p) - \mu(\tau, \mathbf{w}))^2.$$

Rearranging gives

$$((k-1)\alpha_{n,r} + k\alpha_{p,r})Var(\tau, \mathbf{w}^n) - \alpha_{n,r}(\mu(\tau, \mathbf{w}^n) - \mu(\tau, \mathbf{w}))^2$$
$$\leq \alpha_{p,r} \left[ Var(\tau, \mathbf{w}^p) + (\mu(\tau, \mathbf{w}^p) - \mu(\tau, \mathbf{w}))^2 \right]. \tag{33}$$

Let $x = |\mu(\tau, \mathbf{w}^n) - \mu(\tau, \mathbf{w})|$ and $k' = k(\alpha_{n,r} + \alpha_{p,r}) - \alpha_{n,r}$.

$$\frac{Var(\tau, \mathbf{w}^n) + x^2}{Var(\tau, \mathbf{w}^p) + (\frac{\alpha_{n,r}}{\alpha_{p,r}} x)^2} \leq \frac{\alpha_{p,r}}{k'} + (\frac{\alpha_{n,r}}{k'} + 1)\frac{x^2}{Var(\tau, \mathbf{w}^p) + (\frac{\alpha_{n,r}}{\alpha_{p,r}} x)^2}$$

$$\leq \frac{\alpha_{p,r}}{k'} + (\frac{\alpha_{n,r}}{k'} + 1)(\frac{\alpha_{p,r}}{\alpha_{n,r}})^2 \tag{34}$$

$$= \frac{\alpha_{p,r}^2}{\alpha_{n,r}^2} \left( 1 + \frac{\alpha_{n,r}}{\alpha_{p,r}(k - \frac{\alpha_{n,r}}{\alpha_{n,r}+\alpha_{p,r}})} \right)$$

This implies

$$\sum_{i \in S_n} \frac{w_i \tau_i}{\tau_{\max}} \leq \frac{\alpha_{p,r}}{\alpha_{n,r}} \left( 1 + \frac{\alpha_{n,r}}{\alpha_{p,r}(k - \frac{\alpha_{n,r}}{\alpha_{n,r}+\alpha_{p,r}})} \right) \sum_{i \in S_p} \frac{w_i \tau_i}{\tau_{\max}}. \tag{35}$$

Since $s_i \leq \eta$ for $i \in \mathcal{S}_{r+1}$ and $\eta \leq s_i \leq \eta + 1$ for any $i \notin \mathcal{S}_{r+1}$, by the induction claim, we have

$$\eta(1 - \alpha - \alpha_{n,r+1})m \leq \left( \frac{\alpha}{1 - 2\alpha + \frac{\alpha}{1+\eta}} + \frac{1}{k-1} \right) (\eta \alpha_{p,r+1} + (\eta + 1)(\alpha - \alpha_{p,r+1})) \, m. \tag{36}$$

Thus,

$$\alpha_{n,r+1} \geq 1 - \alpha - \left( \frac{\alpha}{1 - 2\alpha + \frac{\alpha}{1+\eta}} + \frac{1}{k-1} \right) \left( \frac{\eta + 1}{\eta} \alpha - \frac{\alpha_{p,r+1}}{\eta} \right). \tag{37}$$

Since

$$\left( \frac{\alpha}{1 - 2\alpha + \frac{\alpha}{1+\eta}} + \frac{1}{k-1} \right) \frac{\eta + 1}{\eta} \leq 1, \tag{38}$$

we have

$$\alpha_{n,r+1} \geq 1 - 2\alpha + \frac{\alpha_{p,r+1}}{\eta + 1}. \tag{39}$$

Since $0 \leq \alpha_{p,r+1} \leq \alpha$, it follows that

$$\frac{\alpha_{n,r+1}}{\alpha_{p,r+1}} \geq \frac{1 - 2\alpha + \frac{\alpha}{\eta+1}}{\alpha}, \tag{40}$$

and

$$\alpha_{n,r+1} \geq 1 - 2\alpha. \tag{41}$$

Finally, since $|S|$ decreases by at least 1 at each round, the algorithm terminates. $\qquad \square$

### C.6 PROOF OF LEMMA 5

*Proof.* Since $\mathbf{w} \in \mathcal{C}_{\beta,\xi,t}$, $\sum_{i \in S_n} w_i \geq \xi$. By Lemma 7, for all $\theta, y$,

$$\left( \sum_{j \in S_n} \frac{w_j}{\sum_{i \in S_n} w_i} F_{j,y}(\theta) - \sum_{j=1}^m w_j F_{j,y}(\theta) \right)^2 \leq \frac{1 - \xi}{\xi} t \tag{42}$$

Similarly,

$$\left( \sum_{j \in S_n} \frac{w_j}{\sum_{i \in S_n} w_i} F_{j,y}(\theta) - \frac{1}{|S_n|} \sum_{j \in S_n} F_{j,y}(\theta) \right)^2$$
$$\leq \frac{1 - \alpha - \beta \xi}{\beta \xi} Var_{\mathcal{H}}(\{\mathcal{W}_i\}_{i \in S_n}) \tag{43}$$

The distribution $D_{\mathbf{w}}$ is the compound sampling such that for each sample in the data set of each worker $i$, the sampling probability is $\frac{w_i}{n \sum_{i=1}^m w_i}$. Therefore,

$$d_{\mathcal{H}}(D, D_{\mathbf{w}}) \leq d_{\mathcal{H}}(\cup_{i \in S_n} \mathcal{W}_i, D_{\mathbf{w}}) + d_{\mathcal{H}}(D, \cup_{i \in S_n} \mathcal{W}_i)$$
$$\leq \sqrt{\frac{1 - \xi}{\xi} t} + \sqrt{\frac{1 - \alpha - \beta \xi}{\beta \xi} Var_{\mathcal{H}}(\{\mathcal{W}_i\}_{i \in S_n}}} \tag{44}$$
$$+ d_{\mathcal{H}}(D, \cup_{i \in S_n} \mathcal{W}_i)$$

$\square$

**Lemma 7** (Property of weighted variance). *For $\mathbf{v} = \{v_i\}_{i=1}^m$ and $\mathbf{w} = \{w_i\}_{i=1}^m$, define the weighted mean and variance as*

$$\mu(\mathbf{v}, \mathbf{w}) = \frac{\sum_{i=1}^m w_i v_i}{\sum_{i=1}^m w_i} \quad and \quad Var(\mathbf{v}, \mathbf{w}) = \frac{1}{\sum_{i=1}^m w_i} \sum_{i=1}^m w_i (v_i - \mu(\mathbf{v}, \mathbf{w}))^2.$$

*Consider a weight vector $\mathbf{w} = \mathbf{w}' + \mathbf{w}''$ with $w_i, w_i', w_i'' \geq 0$, $\sum_{i=1}^m w_i = 1$, and $\xi := \sum_{i=1}^m w_i' \in (0, 1)$. Then*

$$|\mu(\mathbf{v}, \mathbf{w}') - \mu(\mathbf{v}, \mathbf{w})| \leq \sqrt{\frac{1 - \xi}{\xi} Var(\mathbf{v}, \mathbf{w})}. \tag{45}$$

*Proof.* By the variance decomposition formula,

$$\begin{aligned} Var(\mathbf{v}, \mathbf{w}) = \xi \left[ Var(\mathbf{v}, \mathbf{w}') + (\mu(\mathbf{v}, \mathbf{w}') - \mu(\mathbf{v}, \mathbf{w}))^2 \right] \\ + (1 - \xi) \left[ Var(\mathbf{v}, \mathbf{w}'') + (\mu(\mathbf{v}, \mathbf{w}'') - \mu(\mathbf{v}, \mathbf{w}))^2 \right]. \end{aligned} \tag{46}$$

Since variances are nonnegative, we have

$$Var(\mathbf{v}, \mathbf{w}) \geq \xi(\mu(\mathbf{v}, \mathbf{w}') - \mu(\mathbf{v}, \mathbf{w}))^2 + (1 - \xi)(\mu(\mathbf{v}, \mathbf{w}'') - \mu(\mathbf{v}, \mathbf{w}))^2. \tag{47}$$

Using the constraint $\mu(\mathbf{v}, \mathbf{w}) = \xi \mu(\mathbf{v}, \mathbf{w}') + (1 - \xi)\mu(\mathbf{v}, \mathbf{w}'')$, we obtain

$$(1 - \xi)(\mu(\mathbf{v}, \mathbf{w}'') - \mu(\mathbf{v}, \mathbf{w}))^2 = \frac{\xi}{1 - \xi} \xi(\mu(\mathbf{v}, \mathbf{w}') - \mu(\mathbf{v}, \mathbf{w}))^2. \tag{48}$$

Combining these yields

$$Var(\mathbf{v}, \mathbf{w}) \geq \left(1 + \frac{\xi}{1 - \xi}\right) \xi(\mu(\mathbf{v}, \mathbf{w}') - \mu(\mathbf{v}, \mathbf{w}))^2 = \frac{\xi}{1 - \xi}(\mu(\mathbf{v}, \mathbf{w}') - \mu(\mathbf{v}, \mathbf{w}))^2. \tag{49}$$

Taking square roots completes the proof. $\qquad\square$

### C.7 PROPERTY OF $\mathcal{H}$-DIVERGENCE

For completeness, we briefly prove $d_{\mathcal{H}}(D, \{\mathcal{W}_i\}_{i \in S_n}) = \tilde{O}((\frac{d}{nm})^{\frac{1}{2}})$, by the following lemma. Let $n_{tot} = nm$. The lemma characterizes the property of $\mathcal{H}$-divergence between a distribution and $n_{tot}$ samples from the distribution.

**Lemma 8** (Property of $d_{\mathcal{H}}$). *Let $\widehat{D}$ be $n_{tot}$ data sampled independently from the distribution $D$. Then, with probability $1 - \delta$,*

$$d_{\mathcal{H}}(D, \widehat{D}) \leq 8\sqrt{\frac{2d \log(en_{tot}/d)}{n_{tot}}} + 16\sqrt{\frac{2\ln(4/\delta)}{n_{tot}}} \tag{50}$$

*Proof.* Let $(\mathbf{x}_1, y_1), \ldots, (\mathbf{x}_n, y_n)$ be the $n_{tot}$ data samples. Then, by the property of VC-dimension, for any $y'$,

$$\begin{aligned} |\{(I(y_1 = y')I(h(\mathbf{x}_1) = 1), \\ \ldots, \\ I(y_n = y')I(h(\mathbf{x}_n) = 1)) : h \in \mathcal{H}\}| \\ \leq \left(\frac{en_{tot}}{d}\right)^d \end{aligned} \tag{51}$$

Using Theorem 26.5 and 26.8 in Shalev-Shwartz & Ben-David (2014), we obtain that with probability of at least $1 - \delta$, for every $h \in \mathcal{H}$ we have that

$$d_{\mathcal{H}}(D, \widehat{D}) \leq 8\sqrt{\frac{2d \log(en_{tot}/d)}{n_{tot}}} + 16\sqrt{\frac{2\ln(4/\delta)}{n_{tot}}} \tag{52}$$

$$\square$$

## C.8    PROOF OF LEMMA 6

*Proof.* By the definition of the learning error,

$$
\begin{aligned}
\epsilon_D(h) =& \mathbb{P}_{(x,y)\sim D}[h(x) = 1, y = -1] \\
&+ \mathbb{P}_{(x,y)\sim D}[h(x) = -1, y = 1]
\end{aligned}
\tag{53}
$$

By the definition of $\mathcal{H}$-divergence,

$$
\begin{aligned}
& d_{\mathcal{H}}\left(D, D'\right) \\
&= \sup_{h\in\mathcal{H},\mathbf{a}\in\triangle^1} \left| \sum_{y'\in\mathcal{Y}} a_{y'} \left( \mathbb{P}_{(x,y)\sim D}[h(x) = 1, y = y'] - \mathbb{P}_{(x,y)\sim D'}[h(x) = 1, y = y'] \right) \right| \\
&= \sup_{h\in\mathcal{H},\mathbf{a}\in\triangle^1} \left| \sum_{y'\in\mathcal{Y}} a_{y'} \left( \mathbb{P}_{(x,y)\sim D}[h(x) = -1, y = y'] - \mathbb{P}_{(x,y)\sim D'}[h(x) = -1, y = y'] \right) \right|
\end{aligned}
\tag{54}
$$

Therefore,

$$
|\epsilon_{D'}(h) - \epsilon_D(h)| \le 2d_{\mathcal{H}}\left(D, D'\right)
\tag{55}
$$

$\square$

## C.9    PROOF OF THEOREM A.1

Theorem A.1 can be proved in the same way as Theorem 5.1, except the the variance of normal workers is bounded in the following lemma:

**Lemma 9** (Statistic of non-IID normal workers). *Let $\{\mathcal{W}_i\}_{i=0}^m$ be $m'$ data sets that each data set sampled from $\mathcal{C}_\gamma(\mathcal{Z}^n)$. With probability $1 - \delta$,*

$$
Var_{\mathcal{H}}(\{\mathcal{W}_i\}_{i\in S_n}) = O\left( \frac{1}{\gamma} + \frac{1}{n} + \frac{d\ln(mn/d) + \ln(1/\delta)}{mn} + \sqrt{\frac{\ln\left(1/\delta\right)}{2m}} \right)
\tag{56}
$$

*Proof.* Let $\mathbf{q}_i$ be the vector that worker $i$ sampled from $Dir(\mathbf{p})$. Let $D_{\theta,j}$ be the distribution of value $f_{y'}(\theta, x, y) = I(y = y')I(sign(g_\theta(x)) = 1)$, where $(x, y) \sim D_j$. Let $\mu_{\theta,j}$ be the expectation of $D_{\theta,j}$ and $\mu_\theta$ be the $J$-dimension vector that each $j^{th}$ element is $\mu_{\theta,j}$. Then, we can write $F_{i,y'}(\theta) = \frac{1}{n}\sum_{(x,y)\in\mathcal{W}_i} f_{y'}(\theta, x, y)$ such that $f_{y'}(\theta, x, y) \in [0, 1]$ and each $f_{y'}(\theta, x, y)$ is an random variable from $D_\theta(\mathbf{q}_i) = \sum_{j=1}^J q_{i,j} D_{\theta,j}$. Then,

$$
\begin{aligned}
& Var_{\mathcal{H}}(\{\mathcal{W}_i\}_{i\in S_n}) \\
&= \max_{y\subset\mathcal{Y}} \max_{\theta\subset\Theta} \sum_{i=1}^{m'} \frac{1}{m'} \left( F_{i,y}(\theta) - \frac{1}{m'} \sum_{j=1}^{m'} F_{j,y}(\theta) \right)^2 \\
&\le \max_{y\subset\mathcal{Y}} \max_{\theta\subset\Theta} \frac{1}{m'} \sum_{i=1}^{m'} \max_{\mathbf{q}_i} 4(F_{i,y}(\theta) - \mathbb{E}(F_{i,y}(\theta)))^2 \\
&\quad + \max_{y\subset\mathcal{Y}} \max_{\theta\subset\Theta} \frac{1}{m'} \sum_{i=1}^{m'} 2\left( \mathbf{q}_i \cdot \mu_\theta - \mathbf{p} \cdot \mu_\theta \right)^2
\end{aligned}
\tag{57}
$$

Therefore, the variance is bounded by the variance in the IID setting plus the variance induced by the non-IID feature.

For any $\mu$ such that each element $\mu_w \in [0, 1]$, let $\mathbf{q}$ sampled from $Dir(\mathbf{p})$. Since each element pair of vector $\mathbf{q} - \mathbf{p}$ is negatively correlated,

$$
\mathbb{E}_{\mathbf{q}\sim Dir(\mathbf{p})}[(\mathbf{q} \cdot \mu - \mathbf{p} \cdot \mu)^2] \le \sum_{j=1}^J \mu_j^2 \mathbb{E}[(q_j - p_j)^2] \le \frac{1}{\gamma + 1}
\tag{58}
$$

Thus, with probability $1 - \delta$,

$$\max_{y \subset \mathcal{Y}} \max_{\theta \subset \Theta} \frac{1}{m'} \sum_{i=1}^{m'} 2 \left( \mathbf{q}_i \cdot \mu_\theta - \mathbf{p} \cdot \mu_\theta \right)^2 \leq \frac{1}{2(\gamma + 1)} + \sqrt{\frac{\ln(1/\delta)}{2m}} \tag{59}$$

Thus,

$$Var_{\mathcal{H}}(\{\mathcal{W}_i\}_{i \in S_n}) \leq O(\frac{1}{\gamma} + \frac{1}{n} + \frac{d \ln(mn/d) + \ln(1/\delta)}{mn} + \sqrt{\frac{\ln(1/\delta)}{2m}}) \tag{60}$$

$\square$

## D   EXPERIMENT DETAILS

Our experiment is run on an RTX 4090 GPU. In the label-flip attack, the adversary increments each label by one, with 9 wrapping around to 0. For MNIST experiments, we use a dense neural network (DNN) with 2 hidden layers and 128 neurons in each hidden layer. We conduct 5 experiments for each hyperparameter combination. Our algorithm maintains higher test accuracy and lower attack accuracy with Wilcoxon test p-values all below 0.05.

### D.1   VARIANCE MAXIMIZATION PROCESS

The maximization process in Line 5 of Algorithm 2 in conducted by generalized FedAVG as in Algorithm 4.

For the function $F_i(\theta, \mathbf{a})$, we use the sigmoid function $\sigma(z) = \frac{1}{1+e^{-z}}$ to replace the $sign$ function to induce the hypothesis from the model parameter. Therefore, we let $F_i(\theta, \mathbf{a}) = \mathbb{P}_{(x,y) \sim \mathcal{W}_i}[\sum_{y'} a_{y'} I(y = y') \sigma(g_\theta(x))]$ in Line 5 and Line 6 in Algorithm 2.

---

**Algorithm 4:** Variance Maximization: $\theta, \mathbf{a} \leftarrow \arg\max_{\theta, \mathbf{a}} \sum_{i=1}^m w_i \left( F_i(\theta, \mathbf{a}) - \sum_{j=1}^m w_j F_j(\theta, \mathbf{a}) \right)^2$

---

1 **Worker $i$:**
2 If selected by the server, compute

$$grad_i^r \leftarrow \nabla_{\theta_r, \mathbf{a}_r} F_i(\theta_r, \mathbf{a}_r)$$
$$value_i^r \leftarrow F_i(\theta_r, \mathbf{a}_r)$$

  ;
3 Send $grad_i^r$, $value_i^r$ to server;
4 **Server:**
5 **for** $r = 1, 2, \ldots$ **do**
6 $\quad$ Randomly select $k$ workers;
7 $\quad$ Let $\{grad_i^r, value_i^r\}_{i \in [k]}$ be the received gradient and value from the selected $k$ workers;
8 $\quad$ Let $value^r$ be mean of $\{value_i^r\}_{i \in [k]}$;
9 $\quad$ $grad_r \leftarrow \sum_{i \in [k]} (value_i^r - value^r) grad_i^r$;
10 $\quad$ $\theta_{r+1} \leftarrow \theta_r + \eta_r grad_r$;
11 **end**

---

#### D.1.1   COMPLEXITY OF VARIANCE MAXIMIZATION PROCESS

From Algorithm 4, we can see, for the server-side computational complexity, that our computation burden for the server at each round is almost the same as that of FedAVG with $k$ workers selected each round. By contrast, aggregating the gradients with robust mean estimation techniques needs extra optimization processes. For example, for the geometric median, Weiszfeld's iterative algorithm Brimberg (2016) may be needed.

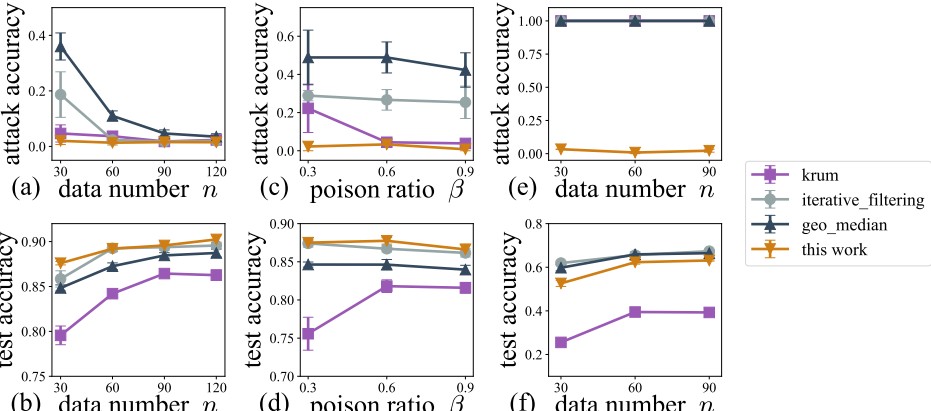

Figure 3: Experimental results. (a-d) are conducted against the backdoor attack. (e,f) are conducted in the CIFAR-10 dataset.

## E    FURTHER EXPERIMENT RESULT

**Evaluation under backdoor attack.** Figure 3(a-d) demonstrates that our algorithm is also effective against the backdoor attack. We reimplement backdoor attacks based on code provided by Bagdasaryan et al.[1] Bagdasaryan & Shmatikov (2021) and the federated learning mechanism and training models based on code provided by Lin et al.[2] Lin et al. (2020). We consider 200 normal workers and 50 workers with poisoned data. In Figure 3(a-b), we consider $n$ samples for each worker, and all data in each poisoned dataset are backdoor data. In Figure 3(c-d), we consider 30 samples for each worker, and $\beta$ proportion of data in each poisoned dataset is backdoor data. Our algorithm maintains higher test accuracy and lower attack accuracy than the baselines in both settings.

**Evaluation under CIFAR-10.** Figure 2(e-f), demonstrates that our algorithm is also effective in the CIFAR-10 dataset. We use VGG13 as the neural network. We consider $n$ samples for each worker, and all data in each poisoned dataset are label-flip data. Figure 2(e) shows that all the baseline algorithms can not defend against the adversary's free ride. The accuracies are close to 1 in terms of the poisoned data. In contrast, our algorithm effectively suppresses the accuracy on the poisoned data.

**Evaluation under the same complexity budget constraint.** Figure 4(a-b) presents a performance comparison between our proposed algorithm and existing methods under the same complexity budget. The horizontal axis denotes the epoch, defined as the computational complexity associated with processing the entire dataset to perform one gradient update. More specifically, the computation complexity for each epoch is $mnd_g$, where $m$ is the number of workers, $n$ is the number of data samples for each worker, and $d_g$ is the dimension of each gradient. The communication complexity is $nd_g$, since each worker computes the gradient on each of its local data samples before submitting gradients to the server in this experiment. Since our algorithm has two phases, we assign the complexity budget to the two phases proportionally, with 40% going to the first and 60% to the second. For example, with 20 epochs, we use 8 epochs for training the discriminator model and 16 epochs for training the target model. Our algorithm achieves the highest test accuracy and lowest attack accuracy.

**Evaluation under the same complexity privacy constraint.** In Figure 4(c-d), we show the compatibility of our algorithm and the local differential privacy. It presents a performance comparison between our proposed algorithm and existing methods under the same privacy budget. We follow the method in  Abadi et al. (2016) to achieve differential privacy. We add the Gaussian noise to both gradients and model output values, $F_i(\theta, \mathbf{a})$, sent to the server. More specifically, for the output value, we add Gaussian noise $\mathcal{N}(0, \sigma\sqrt{T})$. For the gradient, we first clip the norm of the gradient to 1 and add Gaussian noise $\mathcal{N}(0, \sigma\sqrt{T}I)$, where $T$ is the total number of output values and the

---

[1]https://github.com/ebagdasa/backdoors101

[2]https://github.com/epfml/federated-learning-public-code/tree/master/codes/FedDF-code

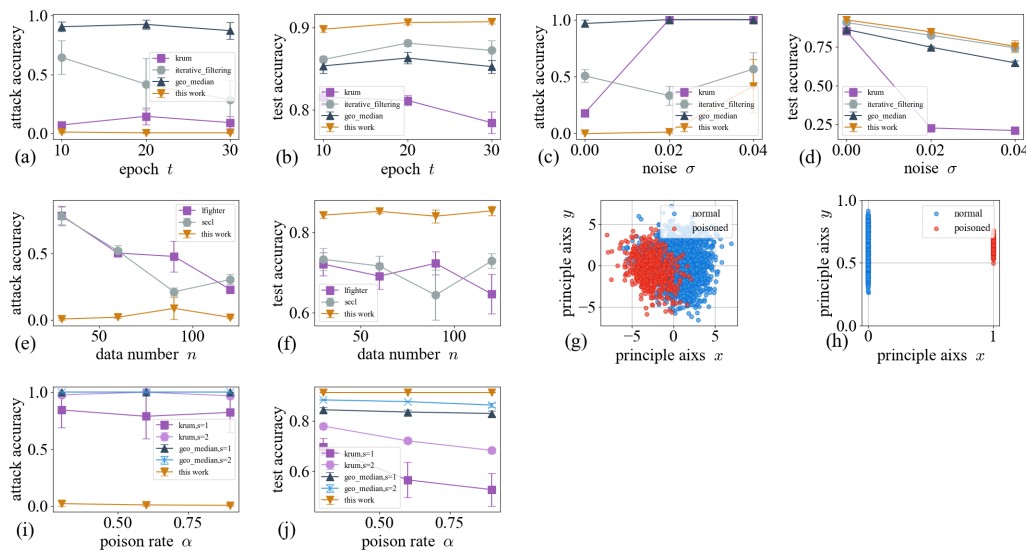

Figure 4: Experimental results. (a-b) are evaluated under the same complexity budget constraint. (c-d) are evaluated under the same privacy budget constraint. (e-f) compares the performance of our algorithm against the label-flipping defenses. (g) illustrates the principal components of gradients calculated by workers. (h) illustrates the principal components of worker datasets in terms of the $\mathcal{H}$-divergence.

gradients sent to the server. Since with the same $\sigma$, each method consumes the same privacy budget, our algorithm achieves the highest test accuracy and lowest attack accuracy under the same privacy budget.

**Comparison with label-flipping defenses.** In Figure 4(e-f), we compare our algorithm with baseline label-flipping defenses Jebreel et al. (2024); Hallaji et al. (2023). LFighter Jebreel et al. (2024) leverages the K-means algorithm to detect workers with poisoned data by clustering the gradients extracted from the last layer of the neural network. Hallaji et al. (2023) incorporates robust training methods to immunize itself against label-flip attacks. Since the algorithm is compatible with various robust training methods, we adopt symmetric cross-entropy learning (SCEL) Wang et al. (2019) as a representative. We use a wider neural network with 1024 neurons in each hidden layer to amplify the differences between algorithms. Our algorithm achieves the highest test accuracy and lowest attack accuracy in a wide neural network.

**Comparison with robust gradient aggregation with bucketing.** In Figure 4(i-j), we compare our algorithm with robust gradient aggregation with bucketing Karimireddy et al. (2022). We consider $s$-bucketing (each bucket having $s$ elements) for Krum and geometric median, with $s = 1$ and $s = 2$. We consider 200 normal workers and 50 workers with poisoned data, so $s$ can at most be 2. We consider the non-IID setting with concentration parameter being 1.0 and 30 data samples per worker. As in Karimireddy et al. (2022), bucketing improves test accuracy in non-iid setting. However, we also observe higher attack accuracy with larger bucket size. Our algorithm achieves the highest test accuracy and lowest attack accuracy in non-iid setting compared to robust gradient aggregation methods with various bucket sizes.

**Intuition that explains our algorithm's effectiveness.** Figure 4(g-h) explains why our discriminator-weighting algorithm is more effective than gradient aggregation. Figure 4(g) illustrates two principal components of gradients calculated by workers, which are two orthogonal projections that maximize the variance of gradients in Euclidean distance. Figure 4(h) illustrates two principal components that maximize the variance of workers' datasets defined in Definition 4.1. Each scatter point represents a worker. The relative distance between the two points illustrates the $\mathcal{H}$-divergence between the two workers' datasets. More specifically, the x-axis (y-axis) of the point

corresponding to the worker $i$ is $F_i(\theta, \mathbf{a})$, where $(\theta, \mathbf{a})$ is obtained in Algorithm 2 Line 5 in round $r = 1$ ($r = 2$).

We can see that it is hard to differentiate the normal datasets from the poisoned datasets from the gradients in Figure 4(g). However, it is easy to differentiate them through the lens of $\mathcal{H}$-divergence in Figure 4(h), since it directly characterizes the distance between datasets. This is because the gradient is an indirect representative of the dataset. Therefore, gradient aggregation can be less effective compared to our approach.

## F  BACKGROUND: FEDERATED LEARNING AND FEDAVG

We describe the problem formulation of federated learning McMahan et al. (2017). A federated learning system consists of federated learning setting with one parameter server and $m$ worker nodes. Each worker $i$ has a loss function $F_i(\theta)$ calculated from his own dataset, where $\theta$ is the model parameter. Federated learning with weight vector $\mathbf{w} = \{w_i\}_{i=1}^m$ aims to minimize the following function:

$$\min_\theta \left\{ F(\theta) \triangleq \sum_{i=1}^m w_i F_i(\theta) \right\} \tag{61}$$

FedAVG McMahan et al. (2017); Chen et al. (2017) is a basic federated learning mechanism in which the worker calculates the gradient of his local dataset and the server collects the gradients and updates the model with a weighted average of the gradients. The mechanism is described in Algorithm 3 in Appendix B.

FedAVG can be generalized to minimize the following function:

$$\min_\theta \left\{ F(\theta) \triangleq f(F_1(\theta), \ldots, F_m(\theta)) \right\} \tag{62}$$

For each round $r$, workers calculate the local gradient $g_{i,r}$ and local value $F_i(\theta_r)$. Then, workers send them to the server so the server can update the model as follows:

$$\theta_{r+1} \leftarrow \theta_r - \eta_r \sum_{i=1}^m \frac{\partial f}{\partial F_i(\theta_r)} grad_{i,r} \tag{63}$$

