# OpenReview forum: "Near Optimal Robust Federated Learning Against Data Poisoning Attack"
_ICLR.cc/2026/Conference — ICLR 2026 Poster_

### Official Review · Reviewer_XehN · 2025-10-17

**Soundness:** 3
**Presentation:** 3
**Contribution:** 3
**Rating:** 6
**Confidence:** 3

**Summary:**

The paper studies robust federated learning (FL) under data poisoning in the regime with many workers \(m\) and few samples per worker \(n\). The authors focus on two goals: minimizing the attack loss (excess error over the attack-free optimum) and bounding a new metric, Effective Poison Rate (EPR), which measures how much of the overall data becomes *effectively* poisoned by the learning pipeline.

They first prove minimax lower bounds on the unavoidable attack loss: in IID settings the loss is \(\Omega(1/\sqrt{n})\); for non-IID modeled with a Dirichlet distribution of concentration \(\gamma\), there is an extra \(\Omega(1/\sqrt{\gamma})\) penalty. Then they propose a two-phase defense. Phase 1 learns trustworthiness weights for workers by training a discriminator to maximize a dataset-variance objective aligned with an \(H\)-divergence view; workers with high deviation get down-weighted. Phase 2 runs weighted FedAvg using these trust weights.

The authors prove upper bounds that match the lower bounds up to logarithmic factors: for IID, attack loss and EPR are \(\tilde{O}\!\left(1/\sqrt{n}+\sqrt{d/(mn)}\right)\); for non-IID, they add \(\tilde{O}(1/\sqrt{\gamma})\). Here \(d\) is a capacity term (e.g., VC dimension). Empirically, on MNIST and CIFAR-10 with label-flip and backdoor attacks, the method tends to improve test accuracy and decrease attack success compared to geometric median, Krum, and iterative filtering, while keeping server-side cost close to standard FedAvg. Overall I feel the paper is promising and quite practical.

**Strengths:**

- Clear objectives: I appreciate the joint focus on attack loss and the new EPR metric.
- Matching Lower and upper bounds align up to logs in both IID and non-IID cases, and the non-IID penalty is expressed cleanly via \(\gamma\).
-The algorithm and results are novel to my knowledge. Using a discriminator on client datasets (outputs) to drive weights avoids heavy dependence on high-dimensional gradient aggregation.
- Practical cost: Communication and compute remain similar to FedAvg; the method seems easy to plug into existing FL pipelines.

**Weaknesses:**

- Baselines are mostly classic robust aggregators. Including more recent FL defenses (NNM, Bucketing, ...) and also poisoning specific defenses (see questions below) would give a fairer picture of current SOTA.


- Many results rely on \(\alpha<1/3\). Whereas most other works only require a majority of honest clients. is there a fundamental reason for this?

- Experiments mainly test label-flip and standard backdoor. Stronger or adaptive poisoning (e.g., clean-label poisoning, gradient-aware poisoning including fall of empires and a liitle is enough) would strengthen the empirical story.

**Questions:**

The following two papers seem very related
   - An Equivalence Between Data Poisoning and Byzantine Gradient Attacks (S. Farhadkhani, R. Guerraoui, L. Hoang, O. Villemaud).
   - On the Relevance of Byzantine Robust Optimization Against Data Poisoning (Sadegh Farhadkhani, Rachid Guerraoui, Nirupam Gupta, Rafael Pinot).
   What is the connection with those results? In particular, how do your EPR notion and your lower/upper bounds relate to the equivalence and to the implications for gradient-robust optimization? Do your results identify regimes where gradient-robustness is provably insufficient but your discriminator-weighting remains effective?


Can your method work under secure aggregation or local differential privacy? If gradients/outputs are obfuscated, can the \(H\)-divergence alignment still be estimated reliably?

In cross-device FL with partial participation, \(\alpha\) can change across rounds. Do your theory and algorithm adapt if the set of corrupted clients varies over time?

---

> ### Author Response · Authors · 2025-11-30
>
> Thank you for your comments. Will improve the paper accordingly.  We clarify some concerns here. We have uploaded the revised manuscript with additional experiment results (in Appendix A), and will discuss these results in the following.
>
> **Q1:**
>
> [Farhadkhani  (2024)] shows equivalence between the two attacks in the setting that workers can only reveal the gradient and when the gradient heterogeneity is a constant. It also shows that a larger gradient heterogeneity results in a larger attack loss in gradient-robust optimization. In our work, we consider the setting where each worker can interact with the server in multiple rounds, and the messages sent to the server are not limited to the gradient.
>
> In addition, since a larger learning model can cause a larger gradient heterogeneity, it consequently results in a larger attack loss in gradient-robust optimization.
> Our work further mitigates the effect of model size on the attack loss.
>
> [Farhadkhani  (2022)] shows equivalence between the two attacks in the setting of personalized learning. In our work, we consider a traditional federated setting where all workers collaborate to learn the same model. Therefore, the training goal is different, and the equivalence result cannot be transferred.
>
> To identify regimes where gradient-robustness is insufficient, we conduct an experiment for intuition in Appendix A.
>
> Figure 3 (g) illustrates two principal components of gradients calculated by workers, which are two orthogonal projections that maximize the variance of gradients in Euclidean distance.
> Figure 3 (h) illustrates two principal components that maximize the variance of workers’ datasets defined in Definition 4.1. Each scatter point represents a worker. The relative distance between the two points illustrates the H-divergence between the two workers' datasets.
>
> We can see that since the gradient is an indirect characterization of the worker’s dataset, it is hard to differentiate the normal datasets from the poisoned datasets using the gradients in Figure 3 (g). However, it is easy to differentiate them through the lens of H-divergence, which is how we differentiate the poisoned dataset in our discriminator-weighting algorithm.
>
> In addition, our lower bound is proved in the setting where each worker can interact with the server in multiple rounds and can send any message to the server. Our result also shows that in this regime, gradient-robust optimization is insufficient, but our algorithm remains effective.
>
> **Q2 (privacy):**
> We conduct a performance comparison between our algorithm and baseline methods under the same privacy budget to show that our method works under local differential privacy.
> We add the Gaussian noise to both updated values and the gradients to achieve differential privacy.
> Our algorithm achieves the highest test accuracy and lowest attack accuracy under the same privacy budget, as in Figure 3 (c-d) in Appendix A in the revised manuscript.
>
> **Q3 (partial participation):**
> In FL with partial participation, we consider the case that at each round a subset of workers is randomly selected from a total of $(1-\alpha)m$ normal workers and $\alpha m$ workers with poisoned data. Therefore, although the proportion of workers with poisoned data changes at each round, the parameter $\alpha$ does not change across the rounds.
>
> **W1:**
> In Figure 3 (i-j) in Appendix A in the revised manuscript, we compare our algorithm with robust gradient aggregation with bucketing [Karimireddy et al. (2022)].
> We consider $s$-bucketing (each bucket having $s$ elements) for Krum and geometric median, with $s=1$ and $s=2$ in the non-IID setting.
> As in [Karimireddy et al. (2022)], bucketing improves test accuracy in non-iid setting. However, we also observe higher attack accuracy with a larger bucket size.
> Our algorithm achieves the highest test accuracy and lowest attack accuracy in non-iid setting compared to robust gradient aggregation methods with various bucket sizes.
>
> **W2:**
> It is a common problem that high-dimensional robust aggregation has a stricter requirement than honest clients majority, such as the iterative filtering method [Su & Xu (2018)] requires $\alpha \ge 1/4$.
> It is an interesting future work to relax this assumption. We think substituting the variance in Algorithm 2 Line 5, for the fourth momentum may relax this assumption. Happy to expand the discussion here.
>
> **W3:** Thank you for the references. We will conduct corresponding experiments in future versions.

---

### Official Review · Reviewer_73aR · 2025-10-18

**Soundness:** 2
**Presentation:** 3
**Contribution:** 2
**Rating:** 4
**Confidence:** 1

**Summary:**

This paper proposes a mechanism for data poisoning attacks in federated learning that asymptotically achieves the theoretical lower bound of attack loss in both IID and non-IID settings when α<1/3. Additionally, the authors introduce a new metric, EPR, to quantify the effectiveness of data poisoning attacks.

**Strengths:**

1. This paper proposes a mechanism for data poisoning attacks in federated learning that asymptotically achieves the theoretical lower bound of attack loss in both IID and non-IID settings when α<1/3.
2. The authors introduce a new metric, EPR, to quantify the effectiveness of data poisoning attacks.

**Weaknesses:**

**The main contributions of this paper lie in the theoretical domain, an area where I do not feel adequately qualified. Therefore, I will focus my critique primarily on potential weaknesses from a practical perspective.**

1. From a practical perspective, data poisoning attacks are generally considered less threatening to federated learning (FL) systems than model poisoning attacks. The latter can directly manipulate model parameters to make malicious updates resemble benign ones, thereby evading detection more effectively. The paper does not clearly justify why focusing on data poisoning is necessary or impactful given this context.

2. The experimental evaluation includes only four baseline methods, all of which were proposed before 2022. This limited selection raises concerns about the adequacy and fairness of the comparison. Including more recent and stronger baselines would provide a clearer picture of the proposed method’s relative performance and novelty.

3. The proposed method is evaluated only on two small-scale datasets, MNIST and CIFAR-10. While these benchmarks are commonly used, they are insufficient to demonstrate robustness and scalability in realistic FL scenarios. Experiments on larger datasets such as ImageNet would strengthen the empirical claims, and extending the evaluation to other modalities, such as NLP tasks, could further validate the generality of the proposed approach.

**Questions:**

See Weaknesses.

---

> ### Author Response · Authors · 2025-11-30
>
> Thank you for your comments. Will improve the paper accordingly.  We clarify some concerns here. We have uploaded the revised manuscript with additional experiment results (in Appendix A), and will discuss these results in the following.
>
> **W1:** Although data poisoning is less threatening, it is a particular attack that is more practical and easier to conduct [Tolpegin et al. (2020].
> In this work, we study the lower and upper bounds in this scenario. We show that a specified defense for the data poisoning attack is more effective than directly applying the model poisoning defense to defend against the data poisoning attack.
>
> **W2:** We choose the baselines with theoretical proofs and can defend against general attacks. These classic defenses remain widely considered in recent works.
> For more recent baselines, we conduct additional experiments.
>
> In Figure 3 (e-f) in Appendix A in the revised manuscript, we compare our algorithm with baseline label-flipping defenses. LFighter [Jebreel et al. (2024] leverages the K-means algorithm to detect workers with poisoned data by clustering the gradients extracted from the last layer of the neural network. [Hallaji et al. (2023)] incorporates robust training methods to immunize itself against label-flip attacks.
> Our algorithm achieves a higher test accuracy and lower attack accuracy compared to these baselines,
>
> In Figure 3 (i-j) in Appendix A in the revised manuscript, we compare our algorithm with robust gradient aggregation with bucketing [Karimireddy et al. (2022)].
> We consider $s$-bucketing (each bucket having $s$ elements) for Krum and geometric median, with $s=1$ and $s=2$ in the non-IID setting.
> As in [Karimireddy et al. (2022)], bucketing improves test accuracy in the non-iid setting. However, we also observe higher attack accuracy with a larger bucket size.
> Our algorithm achieves the highest test accuracy and lowest attack accuracy in non-iid setting compared to robust gradient aggregation methods with various bucket sizes.
>
> Our algorithm also achieves higher test accuracy and lower attack accuracy, compared to these baselines.
>
> **W3:** Thank you for the advice. We will conduct experiments on larger datasets in future versions.

---

### Official Review · Reviewer_vDhi · 2025-10-28

**Soundness:** 3
**Presentation:** 3
**Contribution:** 3
**Rating:** 6
**Confidence:** 4

**Summary:**

The paper addresses the challenge of defending against data poisoning attacks in federated learning by proposing a novel algorithm. The method first assigns a trustworthiness weight to each worker based on the variance of datasets, and then updates the global model using local gradients weighted by these trustworthiness weights. The authors provide a theoretical analysis of the attack loss, establishing an upper bound that matches the lower bound as the local dataset size approaches infinity. Empirical results further corroborate the theoretical analysis, demonstrating the algorithm’s robustness under data poisoning attacks and its superiority over representative robust aggregators.

**Strengths:**

1. The investigated problem of developing federated learning algorithms robust to data poisoning attacks is important and remains relatively underexplored.

2.  This paper conveys a key message that fine-grained algorithmic designs can defend against data poisoning attacks by exploiting the fact that poisoned workers still strictly follow the algorithmic protocol, rather than relying on coarse-grained robust aggregators.

**Weaknesses:**

1. The lower bounds of the attack loss presented in Theorems 3.1 and 3.2 appear to be not tight, as the upper bounds of the proposed algorithm in Theorems 5.1 and A.1 do not align with them, leaving at least a gap of $\tilde{O}\left(\sqrt{\frac{d}{mn}}\right)$. Could the lower or upper bounds be further improved to close this gap?

2. Related to my above comment, the authors should compare their derived lower bound with that in the case of model poisoning attacks, e.g., $\Omega\left(\frac{\alpha}{\sqrt{n}} + \sqrt{\frac{d_g}{mn}}\right)$ in Yin et al. (2018), and clarify why the lower bound under data poisoning attacks is asymptotically smaller than that under model poisoning attacks.

3. In Equation (8), what is the role of the probability simplex $a$? Furthermore, does the updated value $F_i(\theta, a)$ pose any risk to the privacy of worker $i$? The authors are encouraged to provide further discussion on these points.

4. In line 319, the authors claim that the variance of normal workers’ datasets is small when their datasets are correlated. In my view, this holds only under the i.i.d. setting. Will this statement also hold in the non-i.i.d. case? If so, please provide a detailed explanation, as this is a key insight underlying the proposed algorithm.

5. In Algorithm 2, the server removes a worker when its score exceeds a threshold $\eta$. How is $\eta$ chosen, either theoretically or in the experiments? Furthermore, why is the final size of the trust set $(1-2\alpha)m$ rather than $(1-\alpha)m$, given that only an $\alpha$ fraction of workers are poisoned?


6. The experimental baselines are limited. Several other algorithms defend against data poisoning attacks in federated learning [1] [2]. The authors should include comparisons with these methods in their experiments.

    [1] Jebreel, N. M., Domingo-Ferrer, J., Sánchez, D., \& Blanco-Justicia, A. (2024). Lfighter: Defending against the label-flipping attack in federated learning. Neural Networks, 170, 111-126.

    [2] Hallaji, E., Razavi-Far, R., Saif, M., \& Herrera-Viedma, E. (2023). Label noise analysis meets adversarial training: A defense against label poisoning in federated learning. Knowledge-based systems, 266, 110384.

**Questions:**

My detailed questions are listed in the above section; please refer to it.

---

> ### Author Response · Authors · 2025-11-30
>
> Thank you for your comments. Will improve the paper accordingly. We clarify some concerns here. We have uploaded the revised manuscript with additional experiment results (in Appendix A), and will discuss these results in the following.
>
> **W1:** In this work, we focus on the regime that $m \rightarrow \infty$. In this case, the gap tends to $0$. For finite $m$, it is an interesting future work to improve the lower bound. We think this is possible because the limited number of total training data, i.e., $mn$, provides an additional lower bound on the error of the learned model.
>
> **W2:** The bound in Yin et al. (2018) is $O((\frac{trace(\Sigma_g)}{n})^{\frac{1}{2}}+(\frac{d_g}{mn})^{\frac{1}{2}})$. When $m \rightarrow \infty$, the bound will be $O((\frac{trace(\Sigma_g)}{n})^{\frac{1}{2}})$, where $trace(\Sigma_g)$ is the trace of covariance matrix of gradients. This is much larger than our upper bound $O((\frac{1}{n})^{\frac{1}{2}})$, when $m \rightarrow \infty$.
> This is because $trace(\Sigma_g)$ increases linearly with the number of model parameters $d_g$.
>
> **W3 (simplex $a$):** The probability simplex $a$ is from our generalized H-divergence definition (Definition 2.1). H-divergence [Ben-Davidetal.(2010)] is originally defined for distributions over $\mathcal{X}$.  Definition 2.1 generalizes the definition for distributions over $\mathcal{Z}:=\mathcal{X}\times\mathcal{Y}$.
>
> For two distributions $D$ and $D’$ over $\mathcal{Z}$, the maximization over $a$ in Definition 2.1 refers to choosing the maximum between the divergence of $D_{X|Y=0}$ and $D_{X|Y=0}$, and the divergence of $D_{X|Y=1}$ and $D’_{X|Y=1}$.
>
> **W3 (privacy):** For privacy in our algorithm, the leakage is through the updated values $F_i(\theta,a)$ and the gradients, while previous algorithms are through the gradients. Since the dimension of each gradient is much larger than the value, gradients leak much more private information.
> We also conduct a comparison between our proposed algorithm and existing methods under the same privacy budget.
> We add the Gaussian noise to both updated values and the gradients to achieve differential privacy.
> Our algorithm achieves the highest test accuracy and lowest attack accuracy under the same privacy budget, as in Figure 3 (c-d) in Appendix A in the revised manuscript.
>
> **W4:** This statement does not hold in a strongly non-IID setting.
> Intuitively, as the level of non-IID increases, there will be no relation between workers’ datasets. Then, it will be impossible for any algorithm to identify the poisoned datasets from the normal ones.
>
> Our lower bound result captures this feature. A higher level of non-IIDness represents a smaller concentration parameter $\gamma$, and the lower bound of the attack loss will be larger.
> In addition, our upper bound asymptically matches the lower bound in terms of the concentration parameter $\gamma$, which means our algorithm identifies honest workers to the best of its ability in the non-IID setting.
>
> **W5:** In theory, we choose $\eta\ge \sqrt{3\alpha/(1-3\alpha)}+1$. In our experiments, we set $\eta=1$. This choice was motivated by the theoretical overestimation of $\eta$ and the observation that performance is insensitive to $\eta$.
>
> The final size of the trust set is $(1-2\alpha)m$ because each time we eliminate a poisoned dataset, it is possible that we concurrently eliminate a normal dataset. We employ a conservative estimate because the negative impact of leaving a poisoned dataset is greater than that of mistakenly removing a normal dataset.
>
> **W6:** Thank you for the references. We conduct additional experiments to compare our algorithm with these baseline label-flipping defenses. Our algorithm achieves a higher test accuracy and lower attack accuracy compared to these baselines, as shown in Figure 3 (e-f) in Appendix A in the revised manuscript.

---

### Official Review · Reviewer_pxLN · 2025-10-30

**Soundness:** 3
**Presentation:** 2
**Contribution:** 4
**Rating:** 6
**Confidence:** 4

**Summary:**

This paper addresses the problem of defending against data poisoning attacks in Federated Learning (FL), focusing on the practical scenario where the number of workers $m$ is large, but each has only a small local dataset of size $n$. The authors make three key contributions:

Lower Bounds: They establish minimax lower bounds for the attack loss, proving it is $\Omega(1/\sqrt{n})$ in the IID setting and $\Omega(1/\sqrt{\gamma} + 1/\sqrt{n})$ in a Dirichlet-based non-IID setting (where $\gamma$ is the concentration parameter).

Novel Algorithm: They propose a two-phase algorithm. The core innovation is a trustworthiness weight update phase that trains a discriminator model to maximize a notion of dataset variance, allowing the server to assign weights to workers and effectively identify those with poisoned data. This is fundamentally different from prior robust gradient aggregation methods.

New Metric and Upper Bounds: They introduce a new robustness metric, the Effective Poison Rate (EPR), to quantify the adversary's gain. Their algorithm achieves upper bounds for both attack loss and EPR of $\tilde{O}(1/\sqrt{n} + \sqrt{d/(mn)})$ (IID) and $\tilde{O}(1/\sqrt{\gamma} + 1/\sqrt{n} + \sqrt{d/(mn)})$ (non-IID), which asymptotically match the lower bounds when $m \rightarrow \infty$, demonstrating near-optimality.

**Strengths:**

Novel Defense Strategy: The two-phase algorithm, particularly the trustworthiness weight update phase using a discriminator model, is a creative and well-motivated departure from existing methods.

Theoretical Completeness: Provides a full minimax analysis with matching lower and upper bounds under both IID and non-IID settings, a hallmark of strong theoretical work.

Practical Performance: Empirically demonstrates superior performance over several strong baselines on multiple datasets and attack types, validating the theoretical advantages.

New Metric: The Effective Poison Rate (EPR) is a useful and intuitive metric for quantifying the adversary's success beyond simple test accuracy.

**Weaknesses:**

Computational Overhead: The trustworthiness weight update phase requires training an auxiliary discriminator model, which adds non-trivial computational and communication cost before the main training can begin. While argued to be similar to FedAvg per round, the total cost of this extra phase is not thoroughly compared against baselines.

Strong Assumption on $\alpha$: The theoretical guarantees hold under the assumption that the fraction of malicious workers $\alpha < 1/3$. The performance of the algorithm when this assumption is violated (e.g., $\alpha \geq 1/3$) is not explored empirically or discussed in depth.

Clarity of Presentation: As noted above, the paper is challenging to read. The technical depth is high, but the presentation could do more to guide the reader through the complex ideas and proofs.

**Questions:**

The theoretical analysis and algorithm design assume $\alpha < 1/3$. What is the empirical performance of your method when $\alpha \geq 1/3$? Does the performance degrade gracefully, or is there a sharp breakdown point? Is there a pathway to relax this assumption in future work?

The trustworthiness weight update phase is a pre-training step. How does the computational and communication cost of this phase scale with the number of workers $m$ and the model/discriminator complexity? Could this initial phase become a bottleneck in very large-scale FL systems compared to baselines that perform robust aggregation during the main training?

Your method effectively identifies and down-weights suspicious workers. However, in a strongly non-IID setting, could "honest but unusual" workers (those with rare but valid data distributions) be mistakenly penalized by your variance-maximizing discriminator? How does your theory or experiments account for or mitigate this potential issue?

---

> ### Author Response · Authors · 2025-11-30
>
> Thank you for your comments. Will improve the paper accordingly.  We clarify some concerns here. We have uploaded the revised manuscript with additional experiment results (in Appendix A), and will discuss these results in the following.
>
> **W1&Q2 (Computational Overhead):** To illustrate the effect of the computation complexity on the performance. We conduct experiments to compare our proposed algorithm and existing methods under the same complexity budget in Figure 3 (a-b) in Appendix A in the revised manuscript.
>
> Since our algorithm has two phases, we assign the complexity budget to the two phases proportionally, with 40\% going to the first and 60\% to the second.
> For example, with a budget of $20$ epochs, we use $8$ epochs for training the discriminator model and $16$ epochs for training the target model. By comparison, we use $20$ epochs to train the target model in the baseline algorithms with one training phase.
> Our algorithm achieves the highest test accuracy and lowest attack accuracy under the same complexity budget.
>
> For a larger number of workers, in traditional FedAvg algorithm uses the partial participation mechanism to reduce the computation complexity.
> Since our algorithm is also compatible with the partial participation mechanism, there won’t be a bottleneck in the large-scale learning.
> To show the compatibility, we sample 20% of the workers at each step in our experiment in Figure 3 (a-b), and our algorithm outperforms the baselines.
>
> **W2&Q1 (Assumption on $\alpha$):** We think there will be a sharp breakdown point around $\alpha = 1/3$ if the adversary carefully designs the poisoned datasets. The breakdown point phenomenon is common in robust aggregation. For example, the median method [Blanchard et al. (2017)] has a breakdown point at $\alpha = 1/2$, and the iterative filtering method [Su & Xu (2018)] has a breakdown point at $\alpha = 1/4$.
>
> It is an interesting future work to relax this assumption. We think substituting the variance in Algorithm 2 Line 5, for the fourth momentum may relax this assumption. Happy to expand the discussion here.
>
> **Q3:**  In a strongly non-IID setting, honest workers can be mistakenly penalized.
> Intuitively, as the level of non-IID increases, there will be no relation between workers’ datasets. Then, it will be impossible for any algorithm to identify the poisoned datasets from the normal ones.
>
> Our lower bound result captures this feature. A higher level of non-IIDness represents a smaller concentration parameter $\gamma$, and the lower bound of the attack loss will be larger.
> In addition, our upper bound matches the lower bound in terms of the concentration parameter $\gamma$, which means our algorithm identifies honest workers to the best of its ability.

---

### Meta-Review · Area_Chair_wHKg · 2025-12-08

**Summary:**

This paper addresses data poisoning in Federated Learning by proposing a mechanism that asymptotically meets theoretical lower bounds in both IID and non-IID settings. Three reviewers (pxLN, vDhi, XehN) gave positive ratings, highlighting the strong theoretical analysis, the novel two-phase defense algorithm, and the introduction of the Effective Poison Rate (EPR) metric. Reviewer 73aR gave a borderline rejection due to concerns about practical scale and motivation, but admitted low confidence in the domain.

The authors provided a comprehensive rebuttal, including extensive new experiments that addressed missing baselines, computational overhead, and privacy concerns. Given the solid theoretical contributions and the successful empirical validation provided during the rebuttal, I recommend this paper for acceptance.

**Reviewer Concerns:**

The authors successfully addressed the majority of the reviewers' concerns through their detailed rebuttal and revised manuscript.
Specifically, they alleviated concerns from Reviewers vDhi, 73aR, and XehN regarding insufficient baselines by adding comparisons with recent defenses like LFighter and robust aggregation with bucketing. The concern raised by Reviewer pxLN regarding computational overhead was effectively resolved through new experiments demonstrating superior performance under equal complexity budgets.

While the theoretical assumption of the malicious fraction $\alpha < 1/3$ remains a limitation noted by reviewers, the authors provided a reasonable justification citing breakdown points in high-dimensional robust aggregation, and the lack of large-scale ImageNet experiments is acceptable given the paper's primary focus on learning theory.

**Reviewer Scores:**

All the reviewers may maintain their current scores. The three reviewers who rated the paper as a 6 likely view their score as an accurate reflection of a theoretically sound paper that meets the acceptance bar. The reviewer with the score of 4 may maintain their rating due to a preference for larger-scale empirical validation.

---

### Decision · Program_Chairs · 2026-01-26

Accept (Poster)